# Proteogenomic analysis of air-pollution-associated lung cancer reveals prevention and therapeutic opportunities

**Honglei Zhang[1]\*[†], Chao Liu[2][†], Shuting Wang[3][†], Qing Wang[4][†], Xu Feng[1], Huawei Jiang[5], Li Xiao[4], Chao Luo[3], Lu Zhang[2], Fei Hou[2], Minjun Zhou[6], Zhiyong Deng[2], Heng Li[3], Yong Zhang[7]\*, Xiaosan Su[1]\*, Gaofeng Li[3]\***

[1]Center for Scientific Research, Yunnan University of Chinese Medicine, Kunming, China; [2]Department of Nuclear Medicine, Third Affiliated Hospital of Kunming Medical University, Yunnan Cancer Hospital, Kunming, China; [3]Department of Thoracic Surgery II, Third Affiliated Hospital of Kunming Medical University, Yunnan Cancer Hospital, Kunming, China; [4]Department of Oncology, Qujing First People's Hospital, Kuming, China; [5]Department of Ophthalmology, Second People's Hospital of Yunnan Province, Kunming, China; [6]Department of Family Medicine, Community Health Service Center, Kunming, China; [7]Department of Nephrology, Institutes for Systems Genetics, Frontiers Science Center for Disease Related Molecular Network, West China Hospital, Sichuan University, Chengdu, China

**\*For correspondence:**
hlzhang2014@163.com (HZ);
nankai1989@foxmail.com (YZ);
suxs163@163.com (XS);
ligaofeng@kmmu.edu.cn (GL)

[†]These authors contributed equally to this work

**Abstract** Air pollution significantly impacts lung cancer progression, but there is a lack of a comprehensive molecular characterization of clinical samples associated with air pollution. Here, we performed a proteogenomic analysis of lung adenocarcinoma (LUAD) in 169 female never-smokers from the Xuanwei area (XWLC cohort), where coal smoke is the primary contributor to the high lung cancer incidence. Genomic mutation analysis revealed XWLC as a distinct subtype of LUAD separate from cases associated with smoking or endogenous factors. Mutational signature analysis suggested that Benzo[a]pyrene (BaP) is the major risk factor in XWLC. The BaP-induced mutation hotspot, EGFR-G719X, was present in 20% of XWLC which endowed XWLC with elevated MAPK pathway activations and worse outcomes compared to common *EGFR* mutations. Multi-omics clustering of XWLC identified four clinically relevant subtypes. These subgroups exhibited distinct features in biological processes, genetic alterations, metabolism demands, immune landscape, and radiomic features. Finally, *MAD1 and TPRN* were identified as novel potential therapeutic targets in XWLC. Our study provides a valuable resource for researchers and clinicians to explore prevention and treatment strategies for air-pollution-associated lung cancers.

## eLife Assessment

This **useful** manuscript presents an interesting multi-modal omics analysis of lung adenocarcinoma patients with distinct clinical clusters, mutation hotspots, and potential risk factors identified in cases linked to air pollution. The findings show potential for clinical and therapeutic impact. Some of the conclusions remain **incomplete** as they are based on correlative or suggestive findings, and would benefit from further functional investigation and validating approaches.

## Introduction

Lung cancer is the leading cause of cancer deaths globally (*Bray et al., 2018*). Though the most common cause of lung cancer is tobacco smoking, studies estimate that approximately 25% of lung cancers worldwide occur in individuals who have never smoked (*Parkin et al., 2005*). Recently, lung cancer in never smokers (LCINS) were molecular profiled and new genomic features were revealed (*Chen et al., 2020b*; *Govindan et al., 2012*; *Chen et al., 2020a*; *Zhang et al., 2021a*; *Zhang et al., 2019*; *Xu et al., 2020*; *Gillette et al., 2020*). For now, further stratification of LCINS based on different risk factors would be helpful to reveal the oncogenic mechanisms and develop more targeted therapies. Air pollutants, which can directly affect the pulmonary airway, play crucial roles in promoting lung adenocarcinoma (*Hill et al., 2023*; *Turner, 2020*; *Fajersztajn et al., 2013*). More than 20 environmental and occupational agents are lung carcinogens (*Cogliano et al., 2011*; *IARC Working Group on the Evaluation of Carcinogenic Risks to Humans, 2010*) and amount of studies have been made to investigate molecular mechanisms in tumor progression of air pollution chemicals or components using cell lines or mouse/rat models (*Shi et al., 2017*; *Saravanakumar et al., 2022*; *Abd El-Fattah and Abdelhamid, 2021*). However, a comprehensive molecular characterization of clinical lung cancer samples associated with air pollution is still lacking.

The Xuanwei area has the highest rate of lung cancer in China, and extensive research has established a strong link between lung cancer and exposure to domestic coal smoke (*Mumford et al., 1987*; *Barone-Adesi et al., 2012*; *Chapman et al., 2005*; *Lin et al., 2015*; *Zhang et al., 2016*; *Lan et al., 2002*). Specifically, the etiologic link between smoky coal burning and cancer was epidemiologically established *Mumford et al., 1987*; *Barone-Adesi et al., 2012*, and an association between household stove improvement and lower risk of lung cancer was observed (*Lan et al., 2002*). Moreover, genomic evidence of lung carcinogenesis associated with coal smoke in Xuanwei area, China was provided in our previous study (*Zhang et al., 2016*). Thus, lung cancer in Xuanwei areas exemplifies the ideal disease to study characteristics of lung cancers associated with air pollution. In recent years, the molecular features of Xuanwei lung cancer have been gradually revealed (*Hosgood et al., 2013*; *Wang et al., 2018*; *Wang et al., 2017a*; *Zhang et al., 2021b*). A large sample size with multi-comic molecular profiling is urgently needed to explicit the air pollution chemicals and furthermore propose more targeted therapies.

To better understand the molecular mechanisms and heterogeneity of XWLC and to advance precision medicine, we expanded the sample size of our next-generation sequencing dataset to 169 sample size and performed proteomic and phosphoproteomic profiling on 112 samples. Furthermore, we integrated 107 radiomic features derived from X-ray computed tomography (CT) scans into 115 samples to non-invasively distinguish molecular subtypes. This allowed us to identify potential major risk factors, distinguish the genomic features, and establish clinically relevant molecular subtypes. Our study provides an exceptional resource for future biological, diagnostic, and drug discovery efforts in the study of lung cancer related to air pollution.

## Results

To investigate unique biological features of LUAD associated with air pollution, three previous LUAD datasets related to different carcinogens were used for comparison (*Figure 1a*). CNLC is the subset of lung adenocarcinoma from non-smoking patients in Chinese Human Proteome Project (CNHPP project)(*Xu et al., 2020*) (n=77). TSLC is the subset of lung adenocarcinoma from smoking females in the TCGA-LUAD project (*Cancer Genome Atlas Research, 2014*) (n=168). TNLC is the subset of lung adenocarcinoma from non-smoking females in TCGA-LUAD project (*Cancer Genome Atlas Research, 2014*) (n=102). The clinicopathological characteristics of patients from CNLC, TSLC, and TNLC cohorts were supplied in *Supplementary file 1a and b*.

### Proteogenomic landscape in Xuanwei lung cancer (XWLC)

The present study prospectively collected primary samples of LUAD from 169 never-smoking women from the Xuanwei area in China (*Supplementary file 1c*). The XWLC cohort had a median age of 56 years (*Figure 1b*), and the majority of tissue samples were in the early stages of the disease (145 were stage I/II, and 24 were stage III/IV, *Figure 1c*). A total of 135, 136, 102, and 102 tumor samples were profiled with whole-exome sequencing (WES), RNA-seq, label-free protein quantification, and

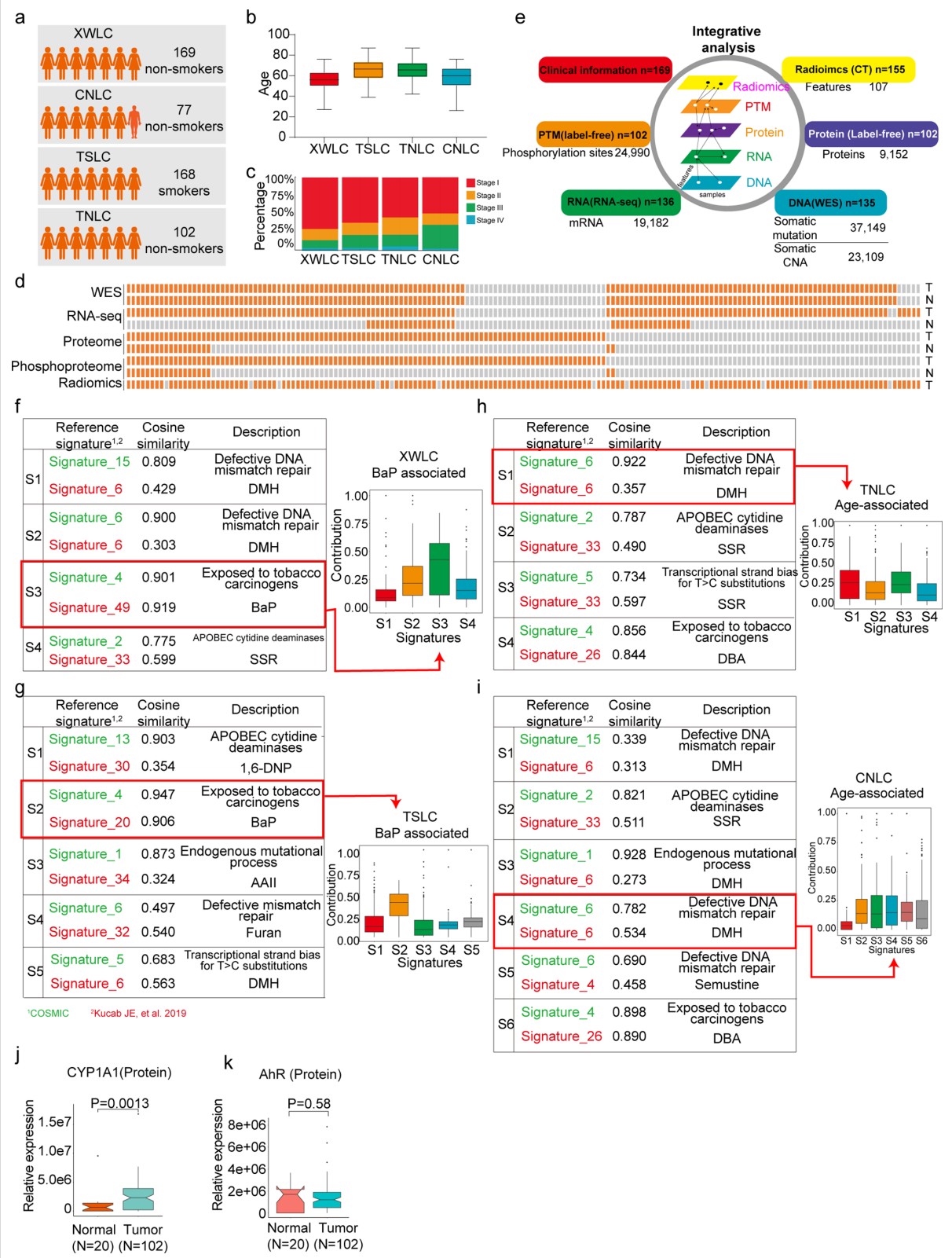

**Figure 1.** Proteogenomic profiling and mutational signatures in Xuanwei lung cancer (XWLC). (**a**) Four cohort datasets used in this study: XWLC (Lung adenocarcinoma from non-smoking females in Xuanwei area), CNLC (subset of lung adenocarcinoma from non-smoking patients in Chinese Human Proteome Project), TSLC (subset of lung adenocarcinoma from smoking females in TCGA-LUAD project), TNLC (subset of lung adenocarcinoma from non-smoking females in TCGA-LUAD project); (**b**) Age distribution of patients at the time of operation in the four cohorts; (**c**) Distribution of tumor

*Figure 1 continued on next page*

*Figure 1 continued*

stages across the cohorts; (**d**) Data availability for the XWLC datasets. Each bar represents a sample, with orange bars indicating data availability and gray bars indicating data unavailability. T, tumor sample. N, Normal tissue; (**e**) Summary of data generated from the XWLC cohort; (**f–i**) Mutational signatures were identified in XWLC (**f**), TSLC (**g**), TNLC (**h**), and CNLC (**i**) cohorts. Cosine similarity analysis of the signatures compared to well-established COMIC signatures (in green) and Kucab et al. signatures (in red). Contribution of signatures in each cohort is provided on the right; (**j–k**) Protein abundance of CYP1A1 (**j**) and AhR (**k**) in tumor and normal samples within the XWLC cohort; Two-tailed Wilcoxon rank sum test used to calculate p-values in (**j–k**).

The online version of this article includes the following figure supplement(s) for figure 1:

**Figure supplement 1.** Experimental workflow and data quality metrics.

**Figure supplement 2.** Identification and profile of de novo mutational signatures in each cohort.

label-free phosphorylation quantification, respectively (*Figure 1d–e* and *Figure 1—figure supplement 1a and b*). Analysis of the WES data from the paired tumor and normal tissue samples revealed 37,149 somatic mutations, including 1797 InDels, 32,972 missense mutations, 2345 nonsense mutations, and 35 nonstop mutations (*Supplementary file 2*). Copy number analysis showed 140,396 gene-level amplifications and 67,605 deletions across 40 cytobands (*Supplementary file 3*). The mRNA-seq data characterized the transcription profiles of 19,182 genes (*Supplementary file 4*). The label-free global proteomics identified 9152 proteins (encoded by 6864 genes) with an average of 6457 proteins per sample (*Supplementary file 5*). The label-free phosphoproteomics identified 24,990 highly reliable phosphosites from 5832 genes with an average of 10,478 phosphosites per sample (*Supplementary file 6*). The quality and reproducibility of the mass spectrometry data were maintained throughout the study (*Figure 1—figure supplement 1c–e*).

## The air pollutant Benzo[a]pyrene (BaP) primarily contributes to the mutation landscape of XWLC

To infer the primary risk factor responsible for the progression of XWLC, we used SomaticSignatures *Gehring et al., 2015* to identify mutational signatures from single nucleotide variants. Mutational signatures were identified in each cohort and a cosine similarity analysis was performed against mutational signatures in COSMIC mutational signatures *Alexandrov et al., 2013b*; *Alexandrov et al., 2013a* and environmental agents mutational sigantures (*Kucab et al., 2019*) allowing for inference of the underlying causes (*Figure 1f–i* and *Figure 1—figure supplement 2*). Generally, exposure to tobacco smoking carcinogens (COSMIC signature 4) and chemicals such as BaP (Kucab signatures 49 and 20) were identified as the most significant contributing factors in both the XWLC and TSLC cohorts (*Figure 1f and g*). In contrast, defective DNA mismatch repair (COSMIC signature ID: SBS6) was identified as the major contributor in both the TNLC and CNLC cohorts (*Figure 1h and i*), with no potential chemicals identified based on signature similarities. Therefore, the XWLC and TSLC cohorts appear to be more explicitly associated with environmental carcinogens, while the TNLC and CNLC cohorts may be more associated with defective DNA mismatch repair processes. BaP, a representative compound of polycyclic aromatic hydrocarbons (PAHs), is found in both cigarette smoke and coal smoke and is recognized as a major environmental risk factor for lung cancer (*Petit et al., 2019*; *Widziewicz et al., 2017*; *Mangal et al., 2009*). Upon metabolism, BaP forms the carcinogenic metabolite 7β,8α-dihydroxy-9α,10α-epoxy-7,8,9,10-tetrahydrobenzo[a]pyrene (BPDE), which creates DNA adducts leading to mutations and malignant transformations. This process involves two key regulators: *CYP1A1* and *AhR*. CYP1A1 plays a crucial role in BaP epoxidation at the 7,8 positions, which is the most critical step in BPDE formation (*Chung et al., 2007*). AhR is a ligand-activated transcription factor that responds to various chemicals, including chemical carcinogens, and is activated by BaP (*Hidaka et al., 2019*). Accordingly, our results demonstrated significantly higher protein expression of CYP1A1 in tumor samples compared to normal samples (*Figure 1j*). AhR showed higher mRNA expression in tumor samples, with no difference in protein level expression (*Figure 1k*). All these results suggested the involvement of BaP and its metabolite in the development of lung cancer.

Though coal-smoke-related lung cancer (XWLC cohort) and cigarette-smoke-related lung cancer (TSLC cohort) showed similar environmental carcinogens, we found that downstream pathway activation and therapeutic targeted potential showed distinctive features. First, the correlation of genomic mutations between XWLC and TSLC was found to be low (*Figure 2a*). Second, there was a remarkable

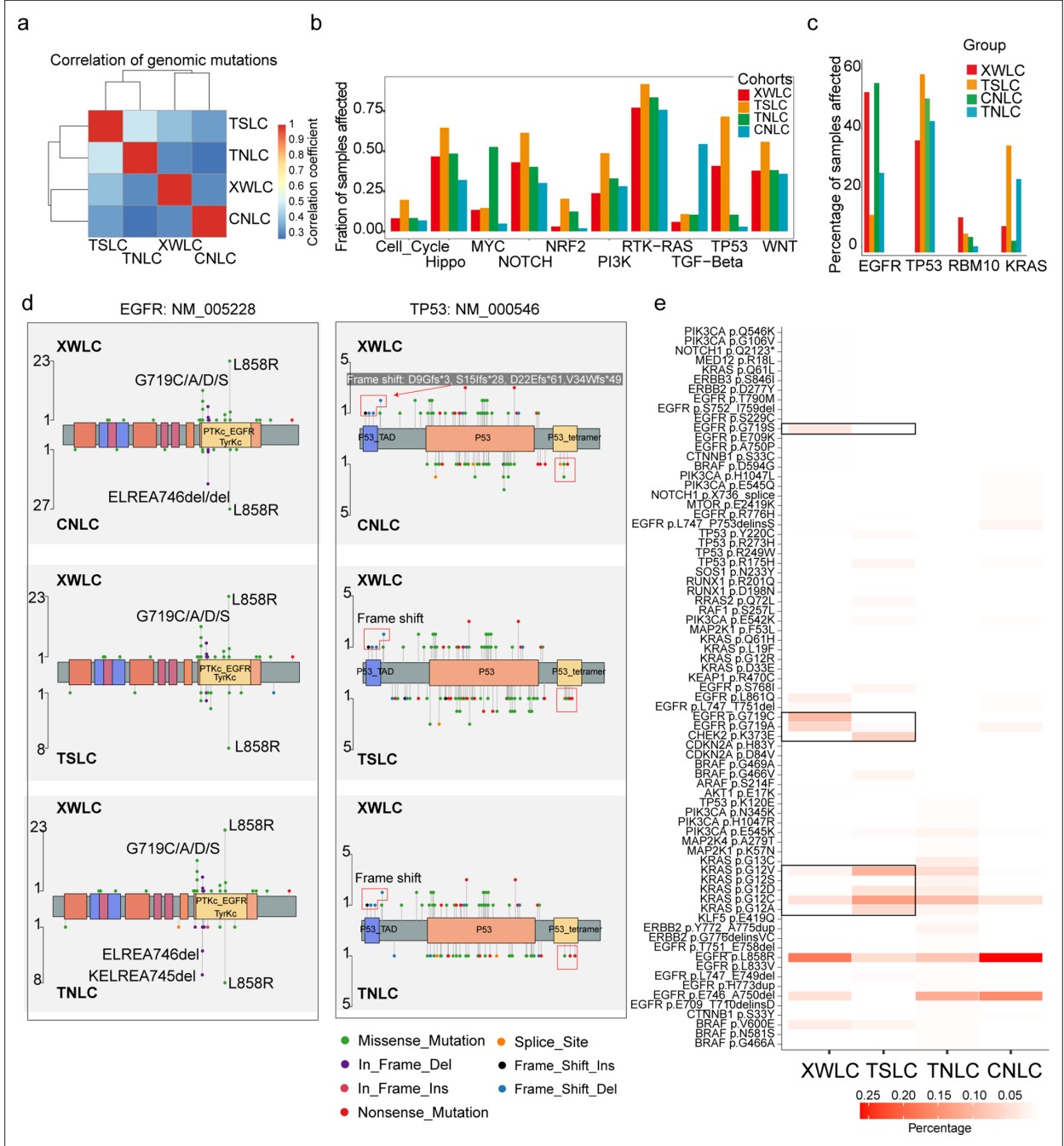

**Figure 2.** Genomic and genetic features in the Xuanwei lung cancer (XWLC) cohort. (**a**) Correlation of genomic mutations among cohorts, determined using the Pearson correlation coefficient; (**b**) Comparison of oncogenic pathways affected by mutations in each cohort; (**c**) Comparison of mutation frequency of four key genes across cohorts; (**d**) Lollipop plot illustrating differences in mutational sites within EGFR (left) and TP53 (right) across XWLC/CNLC, XWLC/TSLC, and XWLC/TNLC pairs; (**e**) Analysis of the percentage of samples with actionable alterations, with a focus on significant variations between XWLC and TSLC cohorts, highlighted by black boxes.

difference in the fraction of samples affected by pathway mutations between the two cohorts (*Figure 2b*). Notably, the TSLC cohort exhibited a higher fraction of samples affected by oncogenic pathways comparing to XWLC cohort. Third, mutation frequencies of top mutated genes (*Zhang et al., 2021b*), such as *EGFR*, *TP53*, *RBM10*, and *KRAS* (*Figure 2c*), as well as the distribution of amino acid changes in *EGFR* and *TP53*, showed noticeable differences between the XWLC and other cohorts (*Figure 2d*). Specifically, the XWLC cohort exhibited a higher mutation rate in G719C/A/D/S within the *EGFR* gene compared to the other three cohorts. For the *TP53* gene, frame shift mutations including

D9Gfs*3 (n=1), S15Ifs*28 (n=1), D22Efs*61 (n=1), and V34Wfs*49 (n=2) were exclusively detected in the XWLC cohort, whereas tetramer domain mutations were only found in the other three cohorts (*Figure 2d*). Finally, there was a noticeable disparity in the percentage of samples with actionable targets among the cohorts. (*Figure 2e*). Actionable targets in the XWLC cohort were mainly focused on *EGFR* mutations including pG719S, pG719C, pG719A, and pL858R, whereas the TSLC cohort had more actionable targets in CHEK2 p.K373E, KRAS p.G12V/D/C/A (*Figure 2e*).

Taken together, we found that the XWLC and TSLC cohorts, which are smoke-related lung adenoma groups, demonstrated distinct etiology compared to the TNLC and CNLC cohorts which may be influenced by endogenous risk factors to a greater extent. Additionally, significant disparities were observed between XWLC and TSLC in terms of downstream pathway activations and specific onco-gene loci. Consequently, we conclude that air pollution-associated lung cancer represents a distinct subtype within LUAD.

## The EGFR-G719X mutation, which is a hotspot associated with BaP exposure, possesses distinctive biological features

Notably, the XWLC cohort displayed a distinguishable mutation pattern in specific *EGFR* mutation sites compared to the other cohorts (*Figure 3a* and *Figure 2d*). In particular, the G719C/A/D/S (G719X) mutation was the most prevalent *EGFR* mutation in the XWLC cohort (20%), while it was rarely found in the other three cohorts (CNLC: 1.9%; TSLC: 1.9%; TNLC: 0) (*Figure 3b*). Notably, we found it was a hot spot associated with BaP exposure (*Figure 3c and d*). Specifically, **GG**C is the 719 codon, the first G can be converted to T (pG719C, n=13) or A (pG719S, n=5), the second G can be converted to A (pG719D, n=1) or C (pG719C, n=8). Thus, pG719C was the most detected mutation type (*Figure 3c*). G>T/C>A transversion can be induced by several compounds such as BaP or dibenz(a,h)anthracene (DBA), unlike other compounds, the tallest peak induced by BaP occurs at GpGpG, reflecting how their DNA adducts are formed principally at $N^2$-guanine (*Kucab et al., 2019*). Our result showed that the most frequently detected pG719C AAchange corresponded to G**G**GC>G**T**GC transversion (*Figure 3d*). Thus, pG719C is a hot spot associated with BaP exposure.

We conducted further investigations into the biological characteristics of samples carrying the G719X mutations. Notably, we observed a moderate to high expression of MAPK signaling components, MAP2K2 (MEK), and MAPK3 (ERK1), in tumors harboring the EGFR-G719X mutation compared to other EGFR statuses (*Figure 3e–h*). Utilizing hallmark capability analysis and RNA-seq-based estimation of immune cell infiltration, we found that tumors with G719X mutations exhibited similarities to those with L858R mutations (*Figure 3—figure supplement 1a–b*). However, patients with G719X mutations were notably younger than those with L858R mutations, indicating a higher occurrence rate of G719X in younger female patients (*Figure 3i*). Analysis of overall survival and progression-free interval (PFI) revealed that patients with the G719X mutation had worse outcomes compared to other EGFR mutation subtypes (*Figure 3j and k*) which was consistent with a previous study (*Watanabe et al., 2014*). Furthermore, there were no significant differences in mutation burden or the number of neoantigens between tumors with G719X mutations and tumors with other *EGFR* mutation statuses (*Figure 3—figure supplement 1c*).

To explore the heterogeneity of signaling pathways activated by different *EGFR* mutation statuses, we conducted a Kinase-Substrate Enrichment Analysis (KSEA) *Wiredja et al., 2017*; *Casado et al., 2013* based on the XWLC phosphoproteomics dataset. Our analysis of the phosphoproteome across various *EGFR* mutation types revealed distinct activation patterns of kinases. Specifically, the G719X mutation was associated with the activation of PRKCZ, CDK2, AURKB, CSNK1A1, CDK4, and HIPK2. The L858R mutation showed activation of PRKCZ, MAPK7, MAPK12, HIPK2, and CSNK2A1. The Exon19del mutation exhibited activation of CHUK, TTK, PRKCZ, PLK1, NEK2, MAP2K2, CDK2, PRKDC, and MAP2K6. Other EGFR mutations were associated with the activation of AURKB, NEK2, TTK, PLK1, PRKACB, and PRKACG. EGFR-WT mutations showed activation of CSNK1E, PRKCZ, AURKB, CDK2, AURKC, CDK1, CSNK1A1, PRKDC, and CSNK2A1 (*Figure 3l*). In *Figure 3—figure supplement 1d*, we provide a list of FDA-approved drugs that target the activated kinases in tumors harboring the G719X mutation. Currently, afatinib is widely regarded as a first-line therapy for patients with the G719X mutation (*Ettinger et al., 2022*; *Janning et al., 2022*; *Yang et al., 2020*; *Cho et al., 2020*). However, reports indicate that 80% of patients with this mutation may develop resistance to afatinib, even in the absence of T790M (*Harada et al., 2019*), underscoring the need for a deeper

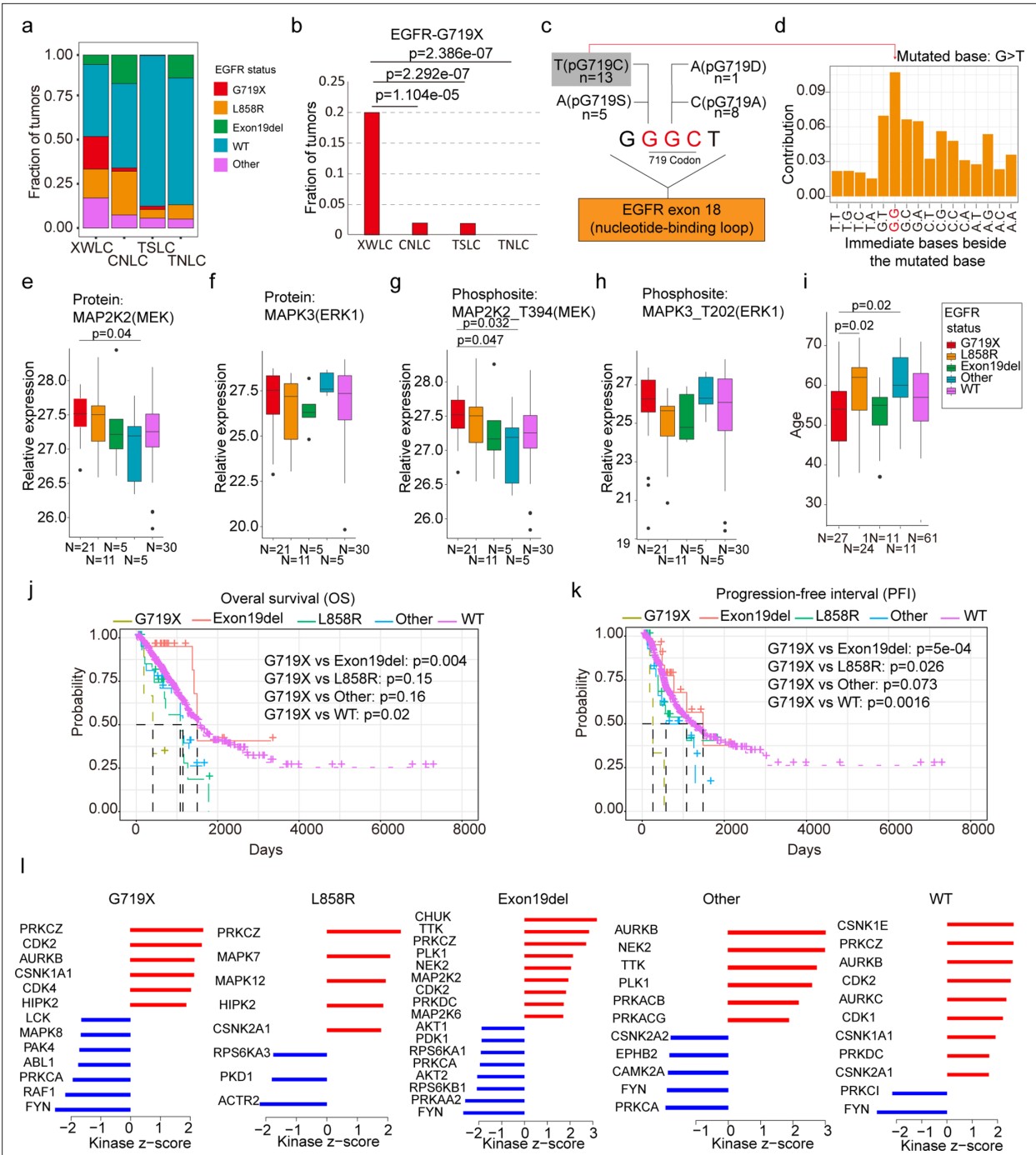

**Figure 3.** EGFR-G719X in the Xuanwei lung cancer (XWLC) cohort. (**a**) Distribution of different EGFR mutation statuses across the four cohorts; (**b**) Comparison of the fraction of G719X mutations across the four cohorts. Two-sided Fisher's test was used to calculate p-values; (**c**) Detailed information on pG719X (pG719/A/D/C/S) mutations in the XWLC cohort. The number of each mutation type is labeled; (**d**) Distribution of nucleotide pairs surrounding the most common G>T transversion site in the XWLC cohort. The x-axis represents the immediate bases surrounding the mutated base. For Benzo[a]pyrene (BaP), the tallest G>T peak occurs at GpGpG; (**e–h**) Comparison of activation levels of key components in the MAPK pathway across different EGFR mutation statuses in the XWLC cohor. N, number of tumor samples containing corresponding EGFR mutation; (**i**) Comparison of patient ages across different EGFR mutation statuses in the XWLC cohort, N, number of tumor samples containing corresponding EGFR mutation; (**j–k**) Presentation of overall survival (OS), (**j**) and progression-free interval (PFI), (**k**) analysis across different EGFR mutation statuses in the TCGA-LUAD cohort, Logrank test was used to calculate p-values; (**l**) Evaluation of kinase activities by KSEA in tumors across different EGFR mutation statuses in the XWLC cohort. The two-tailed Wilcoxon rank sum test was used to calculate p-values in panels (**e–i**).

The online version of this article includes the following figure supplement(s) for figure 3:

**Figure supplement 1.** Comparison of hallmark capability, immune cell infiltration, and mutation burden across different EGFR mutation status.

understanding of the downstream pathways associated with the G719X mutation. Therefore, a promising approach to overcome resistance in tumors with this mutation could involve combining afatinib, which targets activated EGFR, with FDA-approved drugs that specifically target the activated kinases associated with G719X. Therefore, we propose a potential approach to overcoming resistance in tumors with this mutation, which could involve combining afatinib, targeting activated EGFR, with FDA-approved drugs that specifically target the activated kinases associated with G719X.

## Clinically relevant subtyping in XWLC

To uncover the inherent subgroups within air-pollution-associated tumors, we employed unsupervised Consensus Clustering *Wilkerson and Hayes, 2010* on integrated RNA, protein, and phosphoprotein profiles of XWLC tumor samples. This analysis led to the identification of four distinct intrinsic clusters, denoted as MC-I, II, III, and IV (*Figure 4a*, *Figure 4—figure supplement 1a* and Methods). Further survival analysis demonstrated that patients belonging to the MC-IV group exhibited the poorest overall survival compared to the other three subgroups, thus indicating the prognostic potential of multi-omic clustering (*Figure 4b*). Notably, there were no significant differences in clinical features such as age and stage observed among the four subgroups (*Figure 4a*). As CYP1A1 is a key regulator involved in BaP metabolism and has been proven to be highly expressed in tumor samples (*Figure 1j*), we next examined the expression of CYP1A1 among the four subgroups to evaluate their associations with air pollution. Our findings revealed that the MC-II subtype exhibited higher expression of CYP1A1 (*Figure 4c*). Moreover, the MC-II possessed more G719X mutations (MC-1:0.39, MC-II:0.42, MC-III: 0.20, MC-IV: 0.08). Notably, there was a significant correlation between CYP1A1 and EGFR expression (*Figure 4e*), with EGFR being more highly expressed in the MC-II subtype (*Figure 4e*). Collectively, these results indicated that MC-II was more associated with air-pollution.

Through KSEA of the phosphoproteome in tumor samples compared to normal adjacent tissues (NATs), we identified specific kinase activations within the four subgroups. In MC-I samples, kinase activations included PRKDC, PRKCZ, CSNK1A1, NEK2, GSK3A, and ROCK1. MC-II samples showed activations of CDK2, CDK1, AURKA, TTK, CDK6, and CHUK. MC-III samples exhibited activations of AKT1, AKT3, RPS6KB1, CSNK2A1, and PAK2. Finally, CDK2 and ROCK1 were activated in MC-IV samples (*Figure 4f*). Particularly noteworthy is the enrichment of CDK1/2/6 kinases, which regulate cell cycle checkpoints, in the MC-II subtype, indicating its high proliferation capabilities. These findings imply that distinct kinase pathways are activated within each subgroup, suggesting the presence of specific therapeutic targets for each subgroup. Consequently, we proceeded to explore therapeutic strategies for each subgroup as outlined below:

The **MC-IV** subtype exhibited the poorest overall survival compared to the other three subtypes (*Figure 4b*). Given the crucial role of epithelial-mesenchymal transition (EMT) in malignant progression, our first evaluation focused on the EMT process across the four subtypes. We observed higher expression levels of mesenchymal markers such as *VIM*, *FN1*, *TWIST2*, *SNAI2*, *ZEB1*, *ZEB2*, and others in the MC-IV subtype (*Figure 5a*). To comprehensively assess the EMT capability of the MC-IV subtype, we calculated EMT scores using the ssGSEA enrichment method based on protein levels and GSEA hallmark gene set (M5930) (*Liberzon et al., 2015*). The results confirmed the elevated EMT capability of the MC-IV subtype at the protein level (*Figure 5b*). Furthermore, Fibronectin (FN1), an EMT marker that promotes the dissociation, migration, and invasion of epithelial cells, was found to be highly expressed in the MC-IV subtype at the protein level (*Figure 5c*). Additionally, β-Catenin, a key regulator in initiating EMT, was highly expressed in the MC-IV subtype at the protein level (*Figure 5d*). Collectively, our findings demonstrate that the MC-IV subtype is associated with enhanced EMT capability, which may contribute to the high malignancy observed in this subtype.

The **MC-II** subtype demonstrated the second-worst outcome and was found to be more strongly associated with air pollution (*Figure 4*). This subtype exhibited dysregulation of cell cycle processes, including cell division, glycolysis, and cell cycle biological processes (*Figure 4a*). The KSEA analysis revealed that the CDK1 and CDK2 pathways, which are closely linked to cell cycle regulation, were predominantly activated in the MC-II subtype (*Figure 5e*). Consistently, we observed higher expression levels of CDK1 and CDK2 at both the protein and phosphoprotein levels in the MC-II subtype, indicating specific elevation of the G2M phase in the cell cycle (*Figure 5e*). The cell cycle and glycolysis processes are tightly coordinated, allowing cells to synchronize their metabolic state and energy requirements with cell cycle progression to ensure proper cell growth and division (*Vander Heiden*

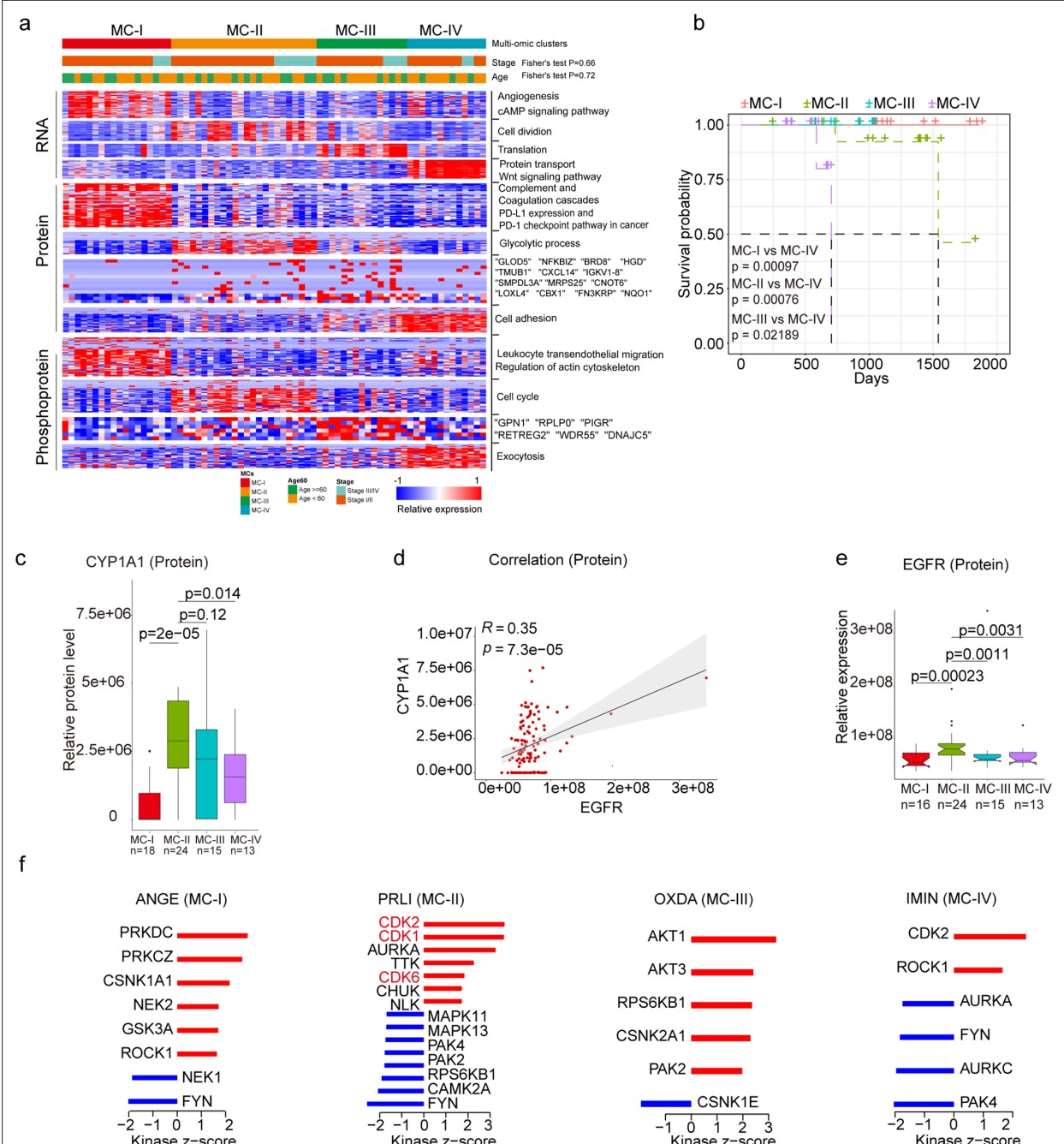

**Figure 4.** Subtyping of Xuanwei lung cancer (XWLC). (**a**) Integrative classification of tumor samples into four ConsensusClusterPlus-derived clusters (MC-I to MC-IV). The heatmap displays the top 50 features, including mRNA transcripts, proteins, and phosphoproteins, for each multi-omic cluster. The features are annotated with representative pathways or genes. If a cluster has fewer than 50 features, all features are shown. If no significant GO biological processes are associated with cluster features, all features are displayed; (**b**) Comparison of overall survival between MC-IV and the other three subtypes; (**c**) Protein abundance comparison of CYCP1A1 across subtypes; (**d**) Protein-level correlation between CYCP1A1 and EGFR; (**e**) Protein-level comparison of EGFR across subtypes; (**f**) Evaluation of kinase activities by Kinase-Substrate Enrichment Analysis (KSEA) in tumors across subtypes in the XWLC cohort. The two-tailed Wilcoxon rank sum test was used to calculate p-values in panels (**c**) and (**e**).

The online version of this article includes the following figure supplement(s) for figure 4:

**Figure supplement 1.** ConsensusClusterPlus clustering in Xuanwei lung cancer (XWLC).

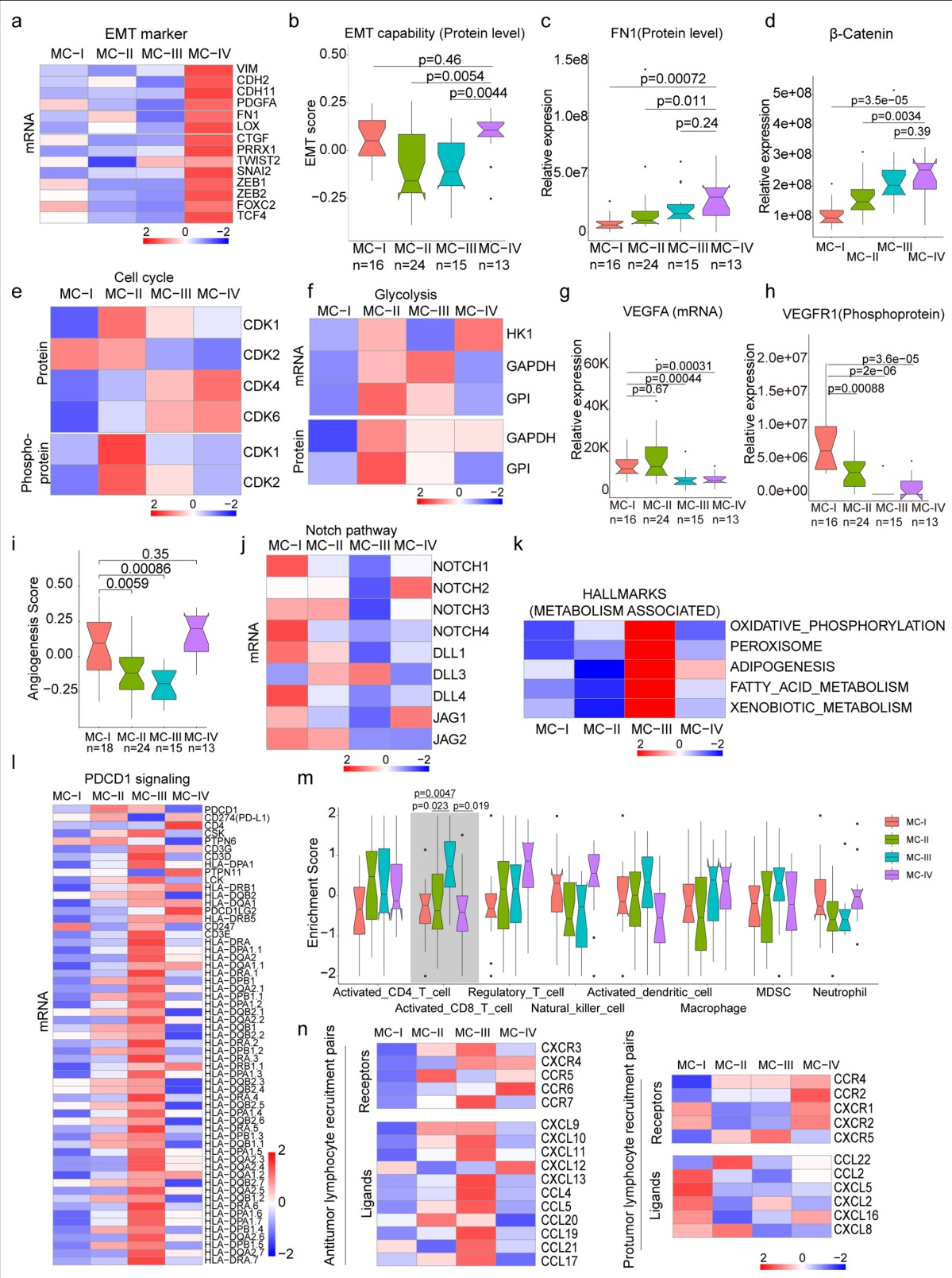

**Figure 5.** Biological and immune features across MC subtypes. (**a**) Relative expression of epithelial-mesenchymal transition (EMT) markers across subtypes; (**b**) EMT scores across subtypes using a gene set derived from MsigDB (M5930); (**c–d**) Protein abundance comparison of FN1 (**c**) and β-Catenin (**d**) across subtypes; (**e**) Protein and phosphoprotein levels of key cell cycle kinases across subtypes; (**f**) Expression of mRNA and protein levels of glycolysis-associated enzymes; (**g**) mRNA expression of VEGFA across subtypes; (**h**) Phosphoprotein abundance of VEGFR1 across subtypes;

*Figure 5 continued on next page*

*Figure 5 continued*

(**i**) Angiogenesis score across subtypes using a gene set derived from MsigDB (Systematic name M5944); (**j**) Expression comparison of key regulators of the Notch pathway across subtypes; (**k**) Metabolism-associated hallmarks across subtypes. Gene sets for oxidative phosphorylation, peroxisome, adipogenesis, fatty acid metabolism, and xenobiotic metabolism were derived from MsigDB hallmark gene sets; (**l**) Expression of PD-1 signaling-associated genes across subtypes. PD-1 signaling-associated genes were derived from MsigDB (Systematic name M18810); (**m**) Immune cell infiltration across subtypes. Gene sets for each immune cell type were derived from a previous study (*Charoentong et al., 2017*); (**n**) Expression of anti-tumor/pro-tumor lymphocyte receptors and ligands across subtypes. The two-tailed Wilcoxon rank sum test was used to calculate p-values in panels **b, c, d, g, h, i, and m**.

The online version of this article includes the following figure supplement(s) for figure 5:

**Figure supplement 1.** KEGG pathview showing the difference across MC subtypes.

---

*et al., 2009*; *DeBerardinis and Thompson, 2012*; *Cairns et al., 2011*). In line with this, we found that key enzymes involved in glycolysis regulation, such as Hexokinase 1 (HK1), Glyceraldehyde-3-phosphate dehydrogenase (GAPDH), and Glyceraldehyde-3-phosphate dehydrogenase-like protein (GPL), were highly expressed in the MC-II subtype (*Figure 5f*). Additionally, the MC-II subtype was enriched with EGFR mutations (MC-II vs. others: 18/24 vs. 51/110; Fisher's exact p=0.013) and TP53 mutations (MC-II vs. others: 14/24 vs. 35/110; Fisher's exact p=0.019), consistent with the characteristic loss of control over cell proliferation. In summary, the MC-II subtype exhibited dysregulated cell cycle processes accompanied by an elevated glycolysis capability, indicating a distinct metabolic and proliferative phenotype.

The **MC-I** subtype exhibited enrichment in various biological processes including angiogenesis, the cAMP signaling pathway, complement and coagulation cascades, PDL1 expression, the PD-1 checkpoint pathway, leukocyte transendothelial migration, and actin cytoskeleton processes (*Figure 4a*). In-depth exploration of key components involved in angiogenesis revealed that vascular endothelial growth factor A (VEGFA), a growth factor crucial for both physiological and pathological angiogenesis, was highly expressed in the MC-I subtype (*Figure 5g*). Additionally, phosphorylation of vascular endothelial growth factor receptor 1 (VEGFR1), a receptor tyrosine kinase essential for angiogenesis and vasculogenesis, was also highly expressed in the MC-I subtype (*Figure 5h*). The angiogenesis scores, calculated using the ssGSEA method based on protein levels and the hallmark gene set (M5944), were relatively high in the MC-I and MC-IV subtypes (*Figure 5i*). Furthermore, the relationship between the Notch signaling pathway and angiogenesis is well-established (*Carmeliet, 2005*). Notch signaling plays a role in multiple aspects of angiogenesis, including endothelial cell sprouting, vessel branching, and vessel maturation (*Pitulescu et al., 2017*; *Gridley, 2007*). In the MC-I subtype, the expression of Notch receptors (Notch1-4) and ligands (DLL1, DLL4, JAG1, and JAG2) was highly elevated, indicating increased activation of Notch signaling (*Figure 5j*). KEGG path view analysis demonstrated that key regulators of the VEGF signaling pathway were highly expressed in the MC-I subtype (*Figure 5—figure supplement 1a*). Therefore, manipulating Notch signaling could potentially serve as a strategy to regulate angiogenesis and control pathological angiogenesis in the MC-I subtype.

The **MC-III** subtype is characterized by the upregulation of various metabolic processes, including oxidative phosphorylation, peroxisome function, adipogenesis, fatty acid metabolism, and xenobiotic metabolism-related processes (*Figure 5k*). Additionally, we conducted further investigations into the immune features across the subtypes. Interestingly, we observed higher expression of genes associated with PD-1 signaling (GSEA, SYSTEMATIC_NAME M18810) in the MC-III subtype (*Figure 5l*). Since PD-1 is primarily expressed on the surface of certain immune cells, particularly activated T cells, we inferred the immune cell infiltration using the ssGSEA method based on immune cell-specific gene sets. We found that activated CD8 + T cells exhibited higher infiltration levels in the MC-III subtype compared to the other three subtypes (*Figure 5m* and *Supplementary file 7*), which may explain the elevated PD-1 signaling in the MC-III subtype. Furthermore, we examined the expression of receptor-ligand pairs involved in both anti-tumor and pro-tumor lymphocyte recruitment. Remarkably, the MC-III subtype exhibited specific high expression of anti-tumor lymphocyte receptors and ligands, while the expression of pro-tumor lymphocyte receptors and ligands was relatively lower (*Figure 5n*). In general, the MC-I subtype showed the reverse expression trend in terms of anti-tumor and pro-tumor receptor-ligand pairs (*Figure 5n*).

In conclusion, our classification of lung adenocarcinoma associated with air pollution resulted in the identification of four subtypes, each exhibiting distinct biological pathway activation and immune

features. The MC-I subtype demonstrated elevated angiogenesis processes, while the MC-II subtype showed a high capacity for cell division and glycolysis. The MC-III subtype exhibited a notable infiltration of CD8 + cells, and the MC-IV subtype was characterized by high EMT capability, which may contribute to its poor outcome. These findings have significant implications for the development of precision treatments for XWLC (*Figure 5—figure supplement 1b*).

## Radiomic features across subtypes

Furthermore, we built a noninvasive method to distinguish MC subtypes with radiomics which entails the extensive quantification of tumor phenotypes by utilizing numerous quantitative image features. In the initial step, we defined 107 quantitative image features that describe various characteristics of tumor phenotypes, including tumor image intensity, size, shape, and texture. These features were derived from X-ray computed tomography (CT) scans of 155 patients with XWLC (Methods). The

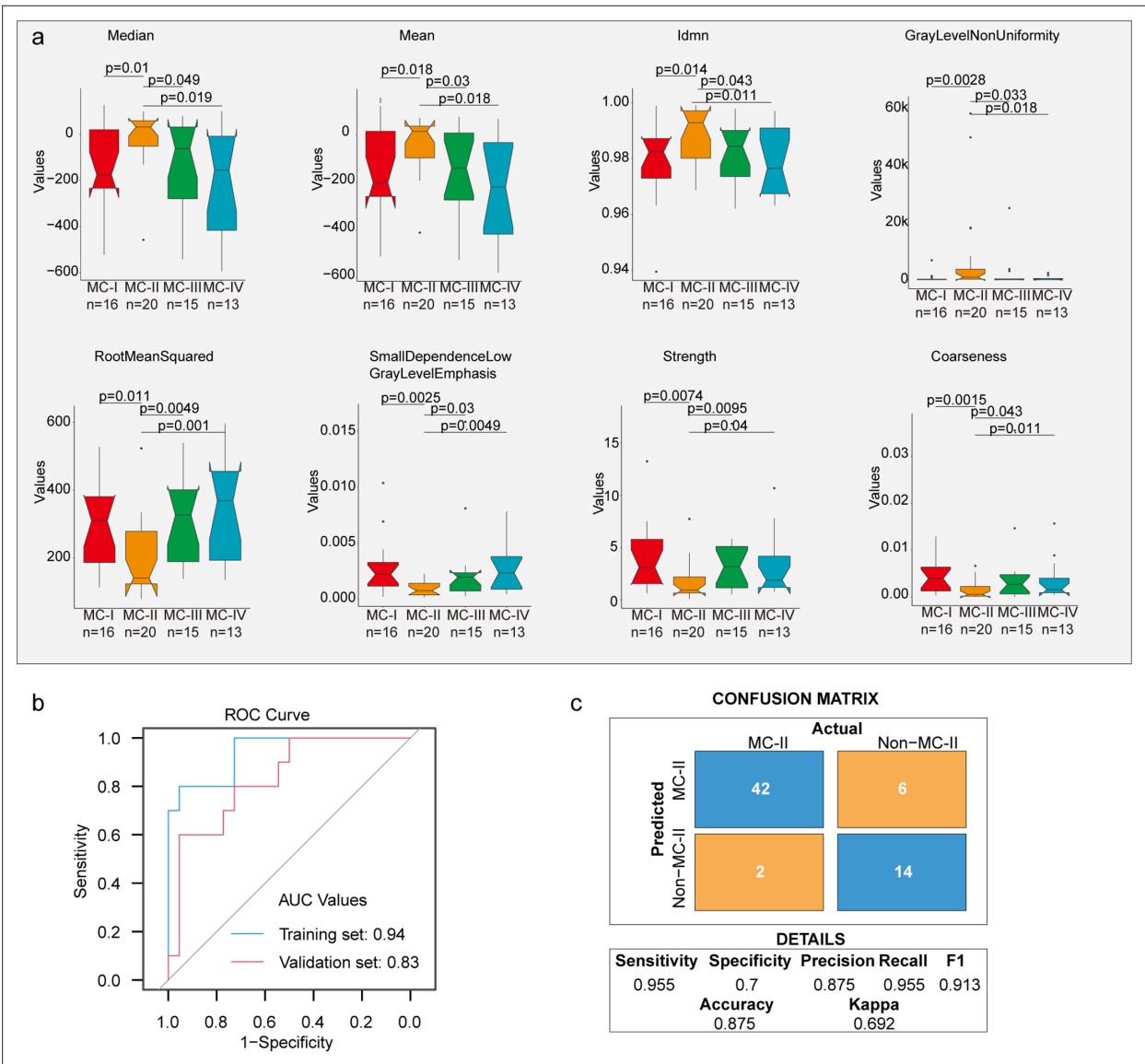

**Figure 6.** Radiomic features across subtypes. (**a**) Eight features showing significant differences between MC-II and the other three subtypes. The Wilcoxon rank sum test was used to calculate the p-values; (**b**) A receiver operating characteristic (ROC) curve was used to evaluate the performance of the radiomic signature in distinguishing MC-II from the other three subtypes; (**c**) Confusion matrix allows visualization of the performance of the algorithm in separating MC-II from other subtypes.

The online version of this article includes the following figure supplement(s) for figure 6:

**Figure supplement 1.** Determination of number of signature features.

baseline characteristics of this cohort can be found in *Supplementary file 8*. Firstly, all features were compared among the four subtypes, and notably, eight features showed significant differences between the MC-II subtype and the other three subtypes (*Figure 6a*). Features such as median and mean reflect the average gray level intensity and Idmn and Gray Level Non-Uniformity measure the variability of gray-level intensity values in the image, with a higher value indicating greater heterogeneity in intensity values. These results suggest a denser and more heterogeneous image in the MC-II subtype. We further established a signature using a multivariate linear regression model with five image features to distinguish MC-II from the other three subgroups (*Figure 6—figure supplement 1*). The performance of the five-feature radiomic signature was validated using the AUC value, which is a generation of the area under the ROC curve. The radiomic signature had an AUC value of 0.94 in the training set and 0.83 in the validation set (*Figure 6b*). The confusion matrix revealed an overall accuracy of 0.875 for sample classification using the signature, indicating proficient performance. However, it exhibited suboptimal performance in terms of false-negative classification (*Figure 6c*). Taken together, we found that MC-II showed a dense image phenotype, which can be noninvasively distinguished using radiomic features.

## Identification of novel targets based on mutation-informed protein-protein interface (PPI) analysis

The integration of genomics and interactomics has enabled the discovery of functional and biological consequences of disease mutations (*Sahni et al., 2015*; *Cheng et al., 2021*). To explore novel targets with the concept, we created PPI networks with structural resolution using missense mutations from the XWLC, CNLC, TSLC, and TNLC cohorts (*Figure 7a* and Methods). OncoPPIs, defined as a significant enrichment of interface mutations in either of the two protein-binding partners across individuals, were identified in each cohort and were provided in *Supplementary file 9*. The OncoPPIs from the four cohorts are named XWLC_oncoPPIs, CNLC_oncoPPIs, TSLC_oncoPPIs, and TNLC_oncoPPIs, respectively (*Figure 7b* and *Figure 7—figure supplement 1a–c*). Initially, the nodes from these four OncoPPIs were subjected to biological process enrichment analysis (*Figure 7—figure supplement 1d*). The analysis revealed that biological processes such as regulation of the mitotic cell cycle, TGF-beta signaling pathway, and immune system were predominantly enriched in the genes related to OncoPPIs. Moreover, the processes disrupted by interface mutations showed a relatively higher similarity between the XWLC and TSLC cohorts (*Figure 7—figure supplement 1d*) suggesting convergent targets or pathways affected by smoke-induced mutations.

To refine the novel targets from XWLC_oncoPPs, we performed molecular dynamics simulations to predict the binding affinity change by the interface-located mutations (Methods). Mitotic Arrest Deficient 1 Like 1 (MAD1), a crucial component of the mitotic spindle-assembly checkpoint (*Wang et al., 2017b*; *Tsukasaki et al., 2001*), forms a tight core complex with MAD2, facilitating the binding of MAD2 to CDC20 (*Yang et al., 2008*), which plays a critical role in sister chromatid separation during the metaphase-anaphase transition (*Sironi et al., 2002*). Specifically, MAD1 Arg558His has been identified as a susceptibility factor for lung cancer *Guo et al., 2010* and colorectal cancer (*Zhong et al., 2015*). Here, we found that MAD1 allele carrying a p.Arg558His substitution may disrupt the interaction between MAD1 and MAD2 (*Figure 7c*). To assess this, we performed molecular dynamics simulations and found that the binding affinity between Arg558His MAD1 and MAD2 was −195.091 kJ/mol and that of the wild-type was −442.712 kJ/mol (*Figure 7c*). Furthermore, on a per-residue basis, the predicted binding affinity (ΔΔG) of Arg558His (*Figure 7—figure supplement 1e*) was projected to increase by 118.319 kJ/mol relative to the wild-type (*Figure 7—figure supplement 1f*), indicating that the substitution of Arg558His in MAD1 perturbs the binding affinity. Thus, our findings suggest that the MAD1 Arg558His attributed to lung cancer progression by disrupting of the interaction between MAD1 and MAD2, which showed potential to be explored as a target.

Notably, we identified TPRN as a novel significantly mutated gene in the XWLC cohort (*Figure 7—figure supplement 1g and h*) whose status was also associated with patients' outcomes (*Figure 7e*). Previous studies have reported that TPRN interacts with PPP1CA (*Ferrar et al., 2012*). Thus, we assessed the binding affinity of the TPRN-PPP1CA complex affected by the mutant variant His550Gln. Our result showed that the binding affinities of the complex were −694.372 kJ/mol and −877.570 kJ/mol in mutant and WT cases, respectively (*Figure 7d*). On a per-residue basis, the predicted ΔΔG of His550Gln compared to the wild-type exhibited an increase of 96.774 kJ/mol (*Figure 7—figure*

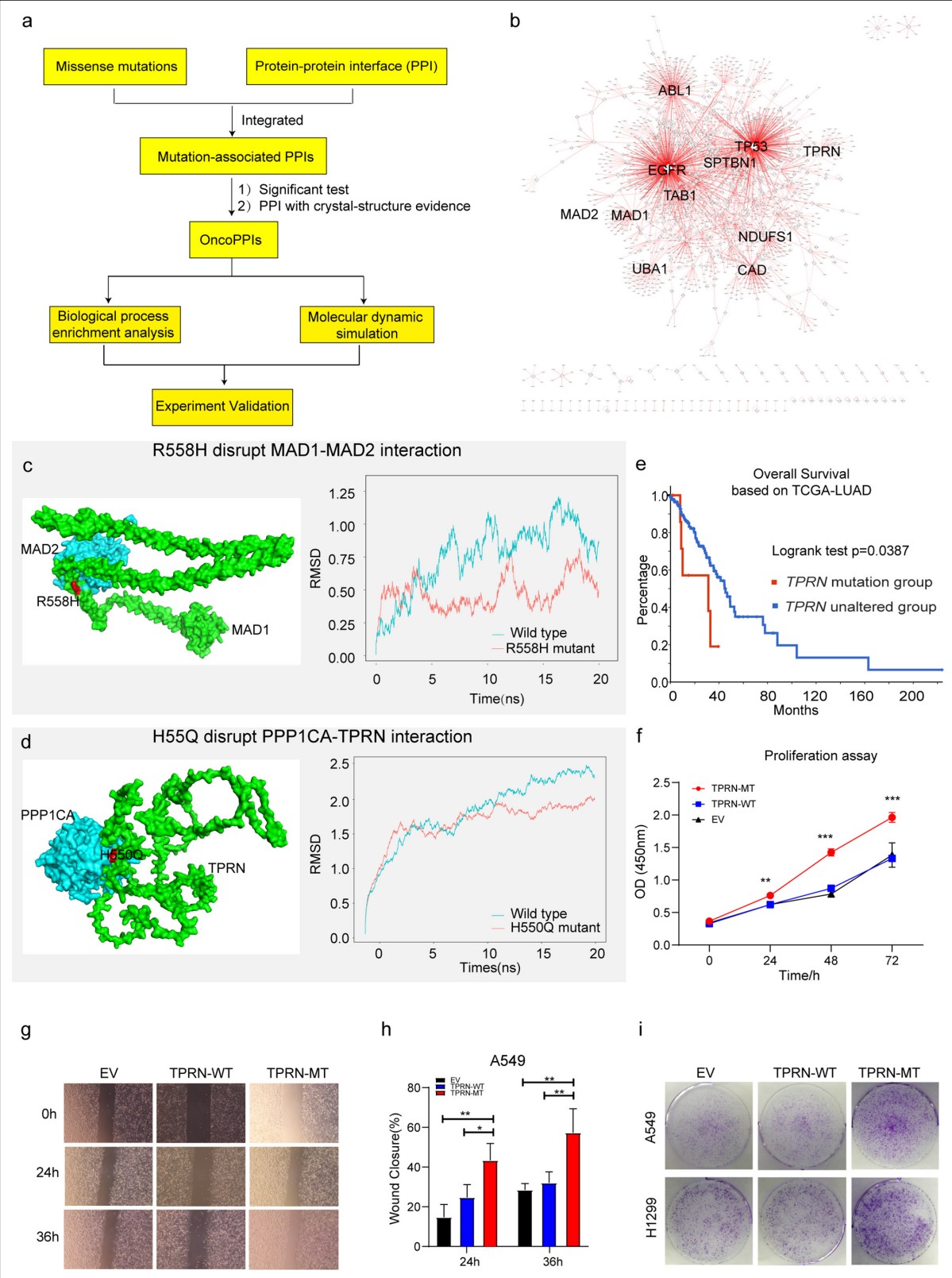

**Figure 7.** Identification of novel targets in Xuanwei lung cancer (XWLC). (**a**) Flow chart showing the integration of mutation-informed PPI analysis, molecular dynamic simulation, and experiment validation to identify novel targets; (**b**) Network visualization of XWLC_oncoPPIs. Edge thickness represents the number of missense mutations at the protein-protein interaction (PPI) interface, while node size indicates connectivity; (**c**) MAD1-MAD2 interaction model and the p.Arg558His mutation at the interface (left). The complex model was generated using Zdock protein docking simulation. The

*Figure 7 continued on next page*

*Figure 7 continued*

right distribution showing root-mean-squared deviation (RMSD) during a 20 ns molecular dynamics simulation of MAD1 wild-type vs. MAD1 p.Arg558His in the complex; (**d**) Model showing the p.His550Gln alteration within the TPRN-PPP1CA complex (left). The right distribution showing root-mean-squared deviation (RMSD) during a 20 ns molecular dynamics simulation for TPRN wild-type vs. TPRN p.His550Gln (H550Q) in the complex; (**e**) Survival analysis of the TPRN mutation group and an unaltered group derived from cbioportal using TCGA-LUAD cohort (https://www.cbioportal.org/); (**f**) CCK8 assay for empty vector (EV), TPRN-WT, and TPRN-MT cell lines in A549 cells which was transfected by EV, TPRN-WT, and TPRN-MT, respectively. (**g**) Transwell assay for EV, TPRN-WT, and TPRN-MT after 24 hr and 36 hr in A549 cells. Magnification was set to 40 x; (**h**) Bar chart showing the statistical results of transwell assay; i. Cell colony assay for EV, TPRN-WT, and TPRN-MT in A549 and H1299 cell lines. All samples were run in triplicate; The two-tailed Wilcoxon rank sum test was used to calculate p-values in **f and h**. *p<0.05; **p<0.01; ***p<0.001.

The online version of this article includes the following figure supplement(s) for figure 7:

**Figure supplement 1.** Networks from oncoPPIs and energy distribution.

**Figure supplement 2.** TPRN p.His550Gln promotes tumor progression in the H1299 cell line.

*supplement 1i–j*). All these results indicated that TPRN His550Gln increase the binding affinity of the TPRN-PPP1CA complex. To investigate the effect of TPRN His550Gln mutation on tumor progression, we examined proliferation and migration capabilities in both A549 and H1299 lung adenocarcinoma cell lines. CCK-8 assay showed significantly enhanced cell growth after transfection of the TPRN mutant allele in both A549 (*Figure 7f*) and H1299 cells (*Figure 7—figure supplement 2a*). Moreover, the wound-healing assay showed that the TPRN mutant cell had achieved enhanced migration capacity (*Figure 7g–h* and *Figure 7—figure supplement 2b–c*). Finally, more cell clones in TPRN His550Gln mutation cells were observed in both TPRN-mutant A549 and H1299 cells (*Figure 7i*). All these results supported that TPRN His550Gln could be explored as a target in XWLC.

Taken together, our integrated analysis of oncoPPIs and molecular dynamics simulations showed the potential to explore novel therapeutic vulnerabilities.

## Discussion

In this study, we conducted proteogenomic and characterized air-pollution-related lung cancers. We found that Benzo[a]pyrene (BaP) influenced the mutation landscape, particularly the EGFR-G719X hotspot found in 20% of cases. This mutation correlated with elevated MAPK pathway activation, worse clinical outcomes, and younger patients. Multi-omics clustering identified four subtypes with unique biological pathways and immune cell patterns. Moreover, our analysis of protein-protein interfaces unveiled novel therapeutic targets. These findings have significant implications for preventing and developing precise treatments for air-pollution-associated lung cancers.

Previously considered uncommon, the EGFR-G719X mutation was detected in only 1–2% of CNLC or TCGA-LUAD cohort samples. Limited knowledge exists from G719X, mostly based on isolated case reports or small series studies (*Chiu et al., 2015*; *Kunishige et al., 2023*; *Yang et al., 2015*; *Massarelli et al., 2013*; *Han et al., 2005*). In vitro experiments using G719X mutant cell lines and patient-derived xenografts (PDX) demonstrated that osimertinib effectively inhibits signaling pathways and cellular growth, leading to sustained tumor growth inhibition (*Floc'h et al., 2020*). However, in silico protein structure analysis suggests that G719 alterations may confer osimertinib resistance due to reduced EGFR binding (*Yang et al., 2018*). Presently, afatinib is proposed as the first-line therapy for G719X mutation patients (*Ettinger et al., 2022*; *Janning et al., 2022*; *Yang et al., 2020*; *Cho et al., 2020*). Unfortunately, 80% of G719X patients develop acquired resistance to afatinib without detecting the T790M mutation40. Hence, further mechanistic studies are warranted for G719X. Our study reveals that the G719X mutation is prevalent in the XWLC cohort, significantly impacting treatment selection. Additionally, the large number of G719X samples allowed us to uncover variations in biology and pathway activation, which may facilitate the development of more precise targeted therapies for these patients.

There is substantial evidence linking lung cancer in the Xuanwei area to coal smoke (*Mumford et al., 1987*; *Barone-Adesi et al., 2012*; *Lin et al., 2015*; *Zhang et al., 2021b*; *Wong et al., 2019*; *Vermeulen et al., 2019*). In addition, we conducted a rat model study that demonstrated the induction of lung cancer by local coal smoke exposure (*Zhang et al., 2021b*). However, the specific chemical compound in coal smoke responsible for causing lung cancer remains largely unknown. Previous research has mainly focused on studying indoor concentrations of airborne particles and

BaP (*Mumford et al., 1987*; *Barone-Adesi et al., 2012*; *Lin et al., 2015*; *Lan et al., 2002*; *Kim et al., 2014*). For instance, studies have shown an association between the concentration of BaP and lung cancer rates across counties (*Mumford et al., 1987*). Moreover, improvements in household stoves have led to reduced exposure to benzopyrene and particulate matter, benefiting people's health (*Lan et al., 2002*; *Kim et al., 2014*). However, these studies primarily relied on epidemiological data, which may be influenced by confounding factors. Mechanistically, Qing Wang showed that BaP induces lung carcinogenesis, characterized by increased inflammatory cytokines, and cell proliferative markers, while decreasing antioxidant levels, and apoptotic protein expression (*Wang et al., 2020*). In our study, we used clinical samples and linked the mutational signatures of XWLC to the chemical compound BaP, which advanced the etiology and mechanism of air-pollution-induced lung cancer. In our study, several limitations must be acknowledged. First, although our multi-omics approach provided a comprehensive analysis of the subtypes and their unique biological pathways, the sample size for each subtype was relatively small. This limitation may affect the robustness of the clustering results and the identified subtype-specific pathways. Larger cohort studies are necessary to confirm these findings and refine the subtype classifications. Second, although our study advanced the understanding of air-pollution-induced lung cancer by using clinical samples, the reliance on epidemiological data in previous studies introduces potential confounding factors. Our findings should be interpreted with caution, and further mechanistic studies are warranted to establish causal relationships more definitively. Third, our in silico analysis suggested a potential approach to drug resistance in G719X mutations. However, these predictions need to be validated through extensive in vitro and in vivo experiments. The reliance on computational models without experimental confirmation may limit the clinical applicability of these findings.

In summary, our proteogenomic analysis of clinical tumor samples provides insights into air-pollution-associated lung cancers, especially those induced by coal smoke, and offers an opportunity to expedite the translation of basic research to more precise diagnosis and treatment in the clinic.

## Materials and methods
### Specimen acquisition
Female patients (n=169) from the Xuanwei area with treatment-naive LUAD were recruited from the Yunnan Cancer Hospital & The Third Affiliated Hospital of Kunming Medical University with written informed consent, with an age range from 27 to 76 (mean = 56). All patients are never-smokers (less than 100 cigarettes in their lifetime which was self-reported smoking status) and also her spouse should be never-smokers if they live together. All tumor specimens were reviewed by pathologists to determine the histological subtype, histological grade, and TNM staging. Majority of the tumors analyzed were stage I or II (n=145) and the remainder were stage III or IV (n=24). Follow-up within this cohort of patients was completed in May 2021, and the median follow-up was conducted at 32 months. Fifteen patients died or experienced relapse or metastatic progression at the time of the last follow-up. Blood samples were also collected from the Yunnan Cancer Hospital & The Third Affiliated Hospital of Kunming Medical University when the females from the Xuanwei area (n=219) or non-Xuanwei areas (n=216) came to the hospital for physical examination in the morning and the serum were kept in –80 °C. Xuanwei areas includes rural counties which high lung cancer incidence such as Laibin, Chang Long, Dai Hai, Shui Wan, Qiao Hong and Long Shuang etc. The study was approved by the ethical committees of Yunnan Cancer Hospital & The Third Hospital of Kunming Medical University.

Methods for Sequencing sample preparation, Whole Exome Sequencing (WES) and quantification, WES library preparation and sequencing, WES data quality control, Reads mapping to the reference sequence, Somatic variant calling, Mutation annotation, Copy number calling, GISTIC and MutSig analysis, RNA sequencing and quantification, RNA library preparation for Transcriptome sequencing, RNA quality control, and Quantification of gene expression level please refer to the previous study (*Zhang et al., 2021b*).

## Label-free mass spectrometry methods

The protocols below for total protein extraction, protein quality test, tryptic digestion, LC-MS/MS Analysis, phosphopeptide enrichment using immobilized metal affinity chromatography, and LC-MS/MS were performed as previously described in depth (*Mertins et al., 2018*).

## Total protein extraction (*Gillette et al., 2020*; *Kachuk et al., 2015*; *Wiśniewski et al., 2009*)

Sample was ground individually in liquid nitrogen and lysed with PASP lysis buffer (100 mM $NH_4HCO_3$, 8 M Urea, pH 8), followed by 5 min of ultrasonication on ice. The lysate was centrifuged at 12,000 *g* for 15 min at 4°C and the supernatant was reduced with 10 mM DTT for 1 hr at 56°C, and subsequently alkylated with sufficient IAM for 1 hr at room temperature in the dark. Then samples were completely mixed with four times the volume of precooled acetone by vortexing and incubated at –20°C for at least 2 hr. Samples were then centrifuged at 12,000 *g* for 15 min at 4°C and the precipitation was collected. After washing with 1 mL cold acetone, the pellet was dissolved by dissolution buffer (8 M Urea, 100 mM TEAB, pH 8.5). For Phosphoproteome, the pellet was dissolved by dissolution buffer, which contained 0.1 M triethylammonium bicarbonate (TEAB, pH 8.5) and 6 M urea.

## Protein quality test

BSA standard protein solution was prepared according to the instructions of the Bradford protein quantitative kit, with gradient concentration ranged from 0 to 0.5 g/L. BSA standard protein solutions and sample solutions with different dilution multiples were added into 96-well plates to fill up the volume to 20 µL, respectively. Each gradient was repeated three times. The plate was added 180 µL G250 dye solution quickly and placed at room temperature for 5 min, the absorbance at 595 nm was detected. The standard curve was drawn with the absorbance of the standard protein solution and the protein concentration of the sample was calculated. 20 µg of the protein sample was loaded to 12% SDS-PAGE gel electrophoresis, wherein the concentrated gel was performed at 80 V for 20 min, and the separation gel was performed at 120 V for 90 min. The gel was stained by coomassie brilliant blue R-250 and decolored until the bands were visualized clearly.

## Trypsin treatment (*Zhang et al., 2016*)

Each protein sample was taken and the volume was made up to 100 µL with DB lysis buffer (8 M Urea, 100 mM TEAB, pH 8.5), trypsin and 100 mM TEAB buffer were added, the sample was mixed and digested at 37 °C for 4 hr. Then trypsin and CaCl2 were added and digested overnight. Formic acid was mixed with the digested sample, adjusted pH under 3, and centrifuged at 12,000 *g* for 5 min at room temperature. The supernatant was slowly loaded to the C18 desalting column, washed with

**Table 1.** Liquid chromatography elution gradient table.

| Time (min) | Flow rate (nL/min) | Mobile phase A (%) | Mobile phase B (%) |
|---|---|---|---|
| 0 | 600 | 94 | 6 |
| 2 | 600 | 90 | 10 |
| 45 | 600 | 70 | 30 |
| 48 | 600 | 65 | 35 |
| 50 | 600 | 50 | 50 |
| 51 | 600 | 0 | 100 |
| 60.5 | 600 | 95 | 5 |
| 61.5 | 600 | 95 | 5 |
| 62 | 600 | 5 | 95 |
| 67 | 600 | 5 | 95 |
| 70 | 600 | 95 | 5 |

washing buffer (0.1% formic acid, 3% acetonitrile) three times, then added elution buffer (0.1% formic acid, 70% acetonitrile). The eluents of each sample were collected and lyophilized.

## Phosphopeptide enrichment using phos-select iron affinity gel for phosphoproteome only

Add binding buffer to dissolve the lyophilized powder, and centrifuge at 12,000 g at 4 °C for 5 min. The supernatant was loaded to the pretreated IMAC-Fe column and incubated at room temperature for 30 min. Then centrifuge at 2000 g for 30 s and wash with washing solution and water each once. Centrifuge at 2000 g for 30 s. Discard the tube, and replace it with a new, clean centrifuge tube. Elute with elution buffer and lyophilized.

## LC-MS/MS analysis for proteome only

Mobile phase A (100% water, 0.1% formic acid) and B solution (80% acetonitrile, 0.1% formic acid) were prepared. The lyophilized powder was dissolved in 10 µL of solution A, centrifuged at 14,000 $g$ for 20 min at 4°C, and 1 µg of the supernatant was injected into a home-made C18 Nano-Trap column (4.5 cm ×75 µm, 3 µm). Peptides were separated in a home-made analytical column (15 cm ×150 µm, 1.9 µm), using a linear gradient elution as listed in *Table 1*. The separated peptides were analyzed by Q Exactive series mass spectrometer (Thermo Fisher), with an ion source of Nanospray Flex (ESI), spray voltage of 2.1 kV, and ion transport capillary temperature of 320°C. Full scan range from $m/z$ 350 to 1500 with resolution of 60,000 (at $m/z$ 200), an automatic gain control (AGC) target value was $3×10^6$ and a maximum ion injection time was 20 ms. The top 40 precursors of the highest abundant in the full scan were selected and fragmented by higher energy collisional dissociation (HCD) and analyzed in MS/MS, where resolution was 15,000 (at $m/z$ 200), the automatic gain control (AGC) target value was $1×10^5$, the maximum ion injection time was 45 ms, a normalized collision energy was set as 27%, an intensity threshold was $2.2×10^4$, and the dynamic exclusion parameter was 20 s. The raw data of MS detection was named as '.raw.'

## LC-MS/MS analysis for phosphoproteome only

Prepare mobile phase A (100% water, 0.1% formic acid) and B (80% acetonitrile, 0.1% formic acid). The lyophilized powder was dissolved in 10 µL solution A and centrifuged at 14,000 g for 20 min at room temperature. 1 µg supernatant was used for detection. Shotgun proteomics analyses were performed using an EASY-nLCTM 1200 UHPLC system (Thermo Fisher) coupled with a Q Exactive HF-X mass spectrometer (Thermo Fisher) operating in the data-dependent acquisition (DDA) mode. 1 µg sample was injected into a home-made C18 Nano-Trap column (2 cm ×75 µm, 3 µm). Peptides were separated in a home-made analytical column (15 cm ×150 µm, 1.9 µm), using a linear gradient elution as listed in *Table 2*. The separated peptides were analyzed by Q Exactive HF-X, with ion source of Nanospray Flex(ESI), spray voltage of 2.3 kV and ion transport capillary temperature of 320 °C. Full scan range from $m/z$ 350–1500 with resolution of 120,000 (at $m/z$ 200), an automatic gain control (AGC) target value was $3×10^6$ and a maximum ion injection time was 80 ms. The 30 most abundant precursor ions from full scan were selected and fragmented by higher energy collisional dissociation (HCD) and analyzed in MS/MS, where resolution was 15,000 (at $m/z$ 200), the automatic gain control (AGC) target value was $5×10^4$ and the maximum ion injection time was 100 ms, a normalized collision

**Table 2.** Liquid chromatography elution gradient table.

| Time(min) | Flow rate (nL/min) | Mobile phase A (%) | Mobile phase B (%) |
|---|---|---|---|
| 0 | 600 | 95 | 5 |
| 2 | 600 | 90 | 10 |
| 112 | 600 | 70 | 30 |
| 117 | 600 | 50 | 50 |
| 118 | 600 | 5 | 95 |
| 123 | 600 | 5 | 95 |

energy was set as 27%, an intensity threshold was $5 \times 10^3$, and the dynamic exclusion parameter was 30 s. The raw data of MS detection was named as '.raw'.

## The identification and quantitation of protein

The all resulting spectra were searched against the UniProt database by the search engines: Proteome Discoverer 2.2 (PD 2.2, Thermo). The search parameters are set as follows: mass tolerance for precursor ion was 10 ppm and mass tolerance for product ion was 0.02 Da. Carbamidomethyl was specified as fixed modifications, Oxidation of methionine (M) was specified as dynamic modification, and acetylation was specified as N-Terminal modification in PD 2.2. A maximum of two missed cleavage sites were allowed.

In order to improve the quality of analysis results, the software PD 2.2 further filtered the retrieval results: Peptide Spectrum Matches (PSMs) with a credibility of more than 99% was identified PSMs. The identified protein contains at least 1 unique peptide. The identified PSMs and protein were retained and performed with FDR no more than 1.0%. The protein quantitation results were statistically analyzed by t-test. The proteins whose quantitation was significantly different between experimental and control groups, ($p<0.05$ and $|log2FC|>1$), were defined as differentially expressed proteins (DEP).

## Mutational signature analysis

Based on the single nucleotide substitution and its' adjacent bases pattern of samples, frequencies of 96 possible mutation types for each sample could be estimated. Non-negative matrix factorization (NMF) algorithm was used to estimate the minimal components that could explain the maximum variance among samples in the R package SomaticSignatures (v2.30.0). Number of signatures was determined by choosing the reflection point in the curves of explained variance and residual sum of squares (RSS). Then each component was compared to mutation patterns of 30 validated cancer signatures reported from the COSMIC database (*Tate et al., 2019*) and of 53 carcinogen signatures reported from *Kucab et al., 2019*. individually to identify cancer-related mutational signatures and carcinogen signatures. Cosine similarity analysis was used to measure the similarity between components and signatures, which ranged from 0 to 1, indicating maximal dissimilarity to maximal similarity. After decomposing the matrix of samples' 96 substitution classes into signatures, the contribution of signatures in each sample could be estimated.

## Correlation of genomic mutations

To calculate the correlation of genomic mutations among four cohorts, all mutations in four cohorts were used and the percentage of mutated samples were correlated. The Pearson method was used.

## Actionable target analysis

Use vcf2maf software to re-annotate maf files. Run the maf2maf.pl file with the parameter + hgvs in the vep calling process to re-annotate HGVS information in the maf files. Use the MafAnnotator. py script from the oncokb-annotator annotation software provided by OncoKB (*Chakravarty et al., 2017*) official to annotate the maf files, add ONCOGENIC information, filter the annotated 'Oncogenic' sites, and use ggplot2 to draw statistics.

## Estimating immune cell populations

RNAseq-derived infiltrating immune cell populations were estimated using the ssGSEA approach (*Barbie et al., 2009*; *Subramanian et al., 2005*). 28 subpopulations of TILs were estimated using prior immune signatures (*Charoentong et al., 2017*). The immune cell populations were: Activated_CD4_T_cell, Activated_CD8_T_cell, Activated_dendritic_cell, CD56bright_natural_killer_cell, Central_memory_CD4_T_cell, Central_memory_CD8_T_cell, Effector_memeory_CD4_T_cell, Effector_memeory_CD8_T_cell, Natural_killer_cell, Natural_killer_T_cell, Type_1_T_helper_cell, Type_17_T_helper_cell, CD56dim_natural_killer_cell, Immature_dendritic_cell, Macrophage, MDSC, Neutrophil, Plasmacytoid_dendritic_cell, Regulatory_T_cell, Type_2_T_helper_cell, Activated_B_cell, Eosinophil, Gamma_delta_T_cell, Immature_B_cell, Mast_cell, Memory_B_cell, Monocyte, T_follicular_helper_cell.

## KSEA analysis

KSEA algorithm was used to estimate the kinase activities by using the ratios of all identified phospho-sites between each EGFR mutation subtype and NAT (*Wiredja et al., 2017*; *Casado et al., 2013*) The kinase activity was predicted using the R package KSEAapp (v0.99.0). Annotations of kinase-substrate relationships were downloaded from the website (https://github.com/casecpb/KSEA/; *Wiredja et al., 2017*; *Casado et al., 2013*). NetworKIN predictions score was included and the minimum was set as 5. The minimum number of substrates a kinase must identify to be included was set as 5. p values less than 0.05 was used in this study.

## Subtyping

The top 2000 most variable features in each data level (RNA transcripts, proteins, phosphoproteins) were collected and subjected to clustering using ConsensusClusterPlus (*Wilkerson and Hayes, 2010*). The top 6000 most variable features were measured by median absolute deviation. We selected 80% item resampling (pItem), 80% gene resampling (pFeature), a maximum evaluated k of 6 so that cluster counts of 2, 3, 4, 5, 6 are evaluated (maxK), 1000 resamplings (reps), and agglomerative hierarchical clustering algorithm (clusterAlg) upon 1- Pearson correlation distances (distance). We determined the number of clusters (k) based on the Consensus Cumulative Distribution Function (CDF) Plot and Delta Area Plot. Specifically, the number of k was determined when CDF reaches an approximate maximum and Delta Area Plot shows no appreciable increase (DOI:https://doi.org/10.1023/A:1023949509487).

## Isolation of signatures in each cluster

To isolate cluster specific highly expressed genes/proteins/phosphoproteins in each cluster, we applied DESeq2 *Love et al., 2014* to calculate significantly up-regulated features by comparing the cluster samples to other cluster samples. FDR <0.05 and fold change >1.2 were used to determine significance.

## Pathway enrichment analysis for cluster signatures

Enriched GO biological processes were extracted from DAVID *Huang et al., 2009* and FDR <0.05 was used to determine the significance.

## Association of clusters to clinical features

To test the association of the resulting clusters to clinical variables we used Fisher's exact test (R function fisher.test) to test for overrepresentation in the set of cluster samples. p<0.05 was used to determine significance.

## Regions of interest (ROI) delineation and inter- and intra-observer reproducibility

All the patients underwent lung cancer examination before biopsy, and CT images were used for radiomics analysis in this study. ROIs were delineated semiautomatically using 3D Slicer software (*Fedorov et al., 2012*). Two radiologists Zhi Li and Chao Liu with 4 and 7 years of experience in lung CT, respectively, were primarily responsible for evaluating the ROIs. The agreements of the ROIs between the radiologists and within the same radiologist represent inter- and intra-observer reproducibility, respectively. The inter- and intra-observer reproducibility of the ROIs and radiomic feature extraction were initially analyzed using the RIDER (*Zhao et al., 2009*) data of 50 randomly selected patients in a blinded fashion by two radiologists. To ensure consistent ROI delineation, one radiologist repeated the ROI delineation twice with an interval of at least 1 month, while another radiologist independently drew the ROIs and generated radiomic features following the same procedure. Intraclass correlation coefficients (ICCs) were used to evaluate the intra- and interobserver agreement in terms of feature extraction. Inter- and intra-observer reproducibility and radiomic feature extraction achieved substantial agreement with ICC >0.80 both among the ROIs from the two radiologists and between the ROIs from the same radiologist.

## Radiomics features

We defined 107 radiomic image features that describe tumour characteristics and can be extracted in an automated way. The features can be divided into three groups: (I) tumour intensity, (II) size and

shape, and (III) texture features. The first group quantified tumour intensity characteristics using first-order statistics, calculated from the histogram of all tumour voxel intensity values. Group 2 consists of features based on the shape of the tumour. Group 3 consists of textual features that are used to quantify intratumour heterogeneity differences in the texture that is observable within the tumour volume. Specifically, they are First Order Statistics (18 features), shape-based (14 features), Gray Level Cooccurence Matrix (24 features), Gray Level Dependence Matrix (GLDM) Features (14 features), Gray Level Run Length Matrix (GLRLM) Features (16 features), Gray Level Size Zone Matrix (GLSZM) Features (16 features), Neighbouring Gray Tone Difference Matrix (NGTDM) Features (5 features). All feature algorithms were implemented with pyradiomics (3.0.1) (*van Griethuysen et al., 2017*).

## Feature selection and radiomics model building

The LASSO method was used to select the most useful predictive features from the training cohort (glmnet R package) (*Friedman et al., 2010*). Tuning parameter ($\lambda$) was selected in the LASSO model by cross-validation for distinguishing XWLC molecular subtypes. Radiomics scores were calculated for each patient using multivariate linear regression (glm R package). The abilities to distinguish XWLC molecular subtypes were assessed using the area under the curve (AUC) of the receiver operator characteristic curve (ROC) via the pROC R package (*Robin et al., 2011*). Radiomics data from patients with multi-omic XWLC subtype and lung CT images were selected to build signatures for distinguishing molecular subtypes inside XWLC. Signature used t0 distinguish MC-II from other subtypes:

$$
\begin{aligned}
\text{Rad-score} = \quad & -1.9203568703 + \\
& 0.0010095229 * \text{original\_firstorder\_Maximum} + \\
& -0.0004598256 * \text{original\_firstorder\_RootMeanSquared} + \\
& 7.4677789388 * \text{original\_glcm\_MaximumProbability} + \\
& 4.5385695277 * \text{original\_glrlm\_GrayLevelNonUniformityNormalized} + \\
& 0.0234377968 * \text{original\_ngtdm\_Busyness}
\end{aligned}
$$

## Radiomics model validation

The radiomics prediction models were validated internally and externally. First, the trained classifiers were assessed by cross-validation via the glmnet R package (*Friedman et al., 2010*). Next, the trained classifiers were further tested in the validation datasets in terms of the AUC of the ROC curve.

## Building mutation-associated protein-protein interactomes

To build structurally resolved mutation-associated protein-protein interaction (PPI) networks based on missense mutations from XWLC, CNLC, TNLC, and TSLC cohort, we downloaded genomic coordinates encoding PPI interfaces from Interactome INSIDER (v.2018.2) (*Meyer et al., 2018*), in which PPI interfaces were compiled for all available 7,135 co-crystal structures and 5386 homology models and 184,605 computationally predicted interactions. We figured out those interfaces carrying missense mutations by mapping mutation sites to PPI interface genomic coordinates using bedtools (v2.25.0) (*Quinlan and Hall, 2010*). Protein UniProt IDs were converted to their gene names based on the UniProt ID mapping tool (*Bateman et al., 2021*). Those PPIs whose protein UniProt IDs conversion failed were discarded. The resulting mutation-associated PPI networks constructed in this way include 27270 PPIs (edges or links) connecting 8756 unique proteins (nodes) in the XWLC cohort, 10706 PPIs connecting 5759 unique proteins in the CNLC cohort, 14829 PPIs connecting 6455 unique proteins in the TNLC cohort and 32157 PPIs connecting 7157 unique proteins in TSLC cohort. All PPIs were experimentally validated PPIs derived from different types of experimental evidence, as described in the original study (*Meyer et al., 2018*).

## Significance test of PPI interface mutations

An oncoPPI is defined as there is significant enrichment in interface mutations in one or the other of the two protein-binding partners across individuals. For each gene $gi$ and its PPI interfaces, we assume that the observed number of cohort-associated mutations for a given interface follows a binomial distribution, binomial ($n$, $pgi$), in which $n$ is the total number of cohort-associated mutations observed in one gene and $pgi$ is the estimated mutation rate for the interface region in one or the other of the

two protein-binding partners of each PPI under the null hypothesis that the region was not recurrently mutated. Using M and N to represent the length of the interface and protein product of gene $gi$, for each interface, we computed the p-value—the probability of observing >k mutations around this interface out of n total mutations observed in this gene—using the following equation:

$$P\left(X \geq k\right) = 1 - P\left(X < k\right) \sum_{x=0}^{k-1} \binom{n}{x} pgi^{x} \left(1 - pgi\right)^{n-x}$$

in which $pgi = \frac{M}{N}$. All p-values were adjusted for multiple testing using the FDR correction. FDR<0.01 was set to determine the significance. In total, we identified 2139 XWLC_oncoPPIs, 1368 CNLC_oncoPPIs, 797 TNLC_oncoPPIs, and 3730 TSLC_oncoPPIs.

## Biological process enrichment in OncoPPI-related genes and term network construction

OncoPPI network was constructed using Cytoscape (v.3.9.1) *Shannon et al., 2003* in four cohorts, respectively. Edge thickness is proportioned to the number of missense mutations at the PPIs interface. Node size is measured by connectivity. Biological process enrichment analysis was performed using Metascape (*Zhou et al., 2019*) based on oncoPPI-related genes. Enrichment term network was built with Metascape (*Zhou et al., 2019*) and major processes were outlined by manual revision.

## Molecular dynamic simulation

We conducted molecular dynamics simulation analysis using X-ray crystallography structures or, in cases where such structures were unavailable, AlphaFold Protein Structure (*Varadi et al., 2022*) predictions. Protein-protein docking was accomplished by ZDOCK (*Pierce et al., 2014*). Mutations were constructed, and incomplete internal sequences or absent terminal regions were modeled in using Swiss-PdbViewer (*Guex and Peitsch, 1997*). Preparation of the system and molecular dynamics simulation were carried out using GROMACS (v.2021.3) *Páll et al., 2020* on the Supercomputing Platform of the Kunming Institute of Zoology. Following a processing step, including the topology for the molecules processed by the pdb2gmx function using CHARMM36 all-atom force field and a water box using TIP3 water molecules with edges at least 10 Å from the protein was added. The system was neutralized to offset the charge on the protein. Energy minimizations of the systems were carried out using the steepest descent until the potential energy was on the order of $10^5$–$10^6$ kJ/mol and the maximum force was no greater than 1000 kJ/mol$^{-1}$ nm$^{-1}$. Following minimization, an NVT and NPT (i.e. constant number of particles, volume and temperature) equilibration step with LINCS constraint algorithm and PME (Particle Mesh Ewald) coulomb type was run using a timestep of 2 fs for 50,000 steps, yielding 0.1 ns of equilibration. Temperature coupling to 300 K was done separately for protein and water/ions using a modified Berendsen thermostat and a 0.1 ps coupling constant. Isotropic pressure coupling to 1 bar was done using a Parrinello–Rahman barostat with a coupling constant of 2.0 ps and compressibility of $4.5 \times 10^{-5}$ bar$^{-1}$. Finally, dynamic simulation was run with no positional restraints for 20 ns using the same 2 fs timestep from equilibration, after which the system was determined by its root-mean-squared deviation (RMSD) to be reasonably well equilibrated.

MM/PBSA energies were calculated over the final 5 ns of each simulation using gmx_MMPBSA (*Valdés-Tresanco et al., 2021*), which can be used for calculating binding free energies of non-covalently bound complexes. Postprocessing and RMSD plots were generated using PyMoL (https://pymol.org/2/, v.2.5.0) and R packages ggplot2.

## Cell culture and TPRN-mutant cell construction

pLV[Exp]-EGFP:T2A:Puro-EF1A (ID: VB220907-1396qay) from VectorBuilder Inc was used in the vector construction. The TPRN His550Gln mutant lentivirus (TPRN-MUT) and its WT lentivirus were constructed. A549 and H1299 cell lines were obtained from Center for Scientific Research, Yunnan University of Chinese Medicine, and separately cultured in RPMI-1640 and DMEM supplemented with 10% fetal bovine serum, and 1% penicillin-streptomycin. The cell culture was maintained in a humidified incubator at 37 °C under 5% CO2. A549 and H1299 cells were then infected with lentivirus.

## Cell viability assay and colony formation assay

Cell viability was assessed by CCK8 (cell counting kit) assay. Briefly, cells were seeded in 96-well plates ($4 \times 10^3$ cells/well) with the indicated treatment. A549 was incubated with CCK8 reagent for 1 hr, while

H1299 was incubated with CCK8 reagent for 1.5 hr at 37 °C. The OD value was measured at 450 nm. The results were statistically analyzed using Tukey's multiple comparisons test with GraphPad Prism software 8.2. For the colony formation assay, A549 and H1299 were separately seeded in a six-well plate ($1 \times 10^3$ cells/well). Twelve days later, the cells were fixed with 4% paraformaldehyde in PBS for 30 min and then stained with crystal violet for 30 min. The dishes were gently washed with PBS three times.

### Wound-healing assay

A549 and H1299 cells were seeded in six-well plates and cultured until 95% confluent. The adherent monolayer cells were scratched using a 200 µl pipette tip and the plate was washed three times. The scratch wounds are marked with dots using labelling pen. The plates were incubated with fresh media containing 2% FBS under 5% CO2 at 37 °C. Images of the scratch wounds were captured using a phase-contrast microscope. Percentage of wound-healing area was calculated using ImageJ as the following formula: (original scratch area - scratch area after healing) (original scratch area)–$1 \times 100\%$, and the results were statistically analyzed using Tukey's multiple comparisons test with GraphPad Prism software 8.2.

### Statistics and reproducibility

Unless specified otherwise, two-sided Fisher's exact testing was used for p- value calculations between two categorical variables, while two-sided Wilcoxon rank sum testing was used between two continuous variables for all figures. Log-rank tests were used for comparison of survival distributions in Kaplan–Meier plots. For multiple testing corrections, FDR corrections were performed unless specified otherwise.

### Acknowledgements

We thank Kesha Yang from Novogene Co., Ltd., Wenfeng Sun from Scale Biomedicine Technology Co., Ltd. (Beijing, China), and Jingli Li from Beijing Qinglian Biotech Co., Ltd. for their help in sequencing and/or bioinformatics analysis. We also thank the 'Open and Shared Public Science and Technology Service Platform of Traditional Chinese Medicine Science and Technology Resources in Yunnan' for providing experimental platforms and equipment. This work was supported by the National Natural Science Foundation (Nos. 82273501, 82360613, 82160343, 82060519); Yunnan Basic Research Program (Nos. 202101AZ070001-002, 202401AS070070, 202301AY070001-106); Yunnan Young and Middle-aged Academic and Technical Leaders Reserve Talents Project (No. 202005AC160048); Yunnan Provincial Department of Education Science Research Fund Project (2024J0246); 'Famous Doctor' Special Project of Ten Thousand People Plan of Yunnan Province (Nos. CZ0096, YNWR-MY-2020–095); and the Medical Leading Talents Training Program of Yunnan Provincial Health Commission (No. L-2019028).

### Additional information

#### Funding

| Funder | Grant reference number | Author |
| --- | --- | --- |
| National Natural Science Foundation of China | 82273501 | Honglei Zhang |
| National Natural Science Foundation of China | 82360613 | Honglei Zhang |
| National Natural Science Foundation of China | 82160343 | Zhiyong Deng |
| National Natural Science Foundation of China | 82060519 | Xiaosan Su |
| Yunnan Basic Research Program | 202101AZ070001-002 | Honglei Zhang |

| Funder | Grant reference number | Author |
|---|---|---|
| Yunnan Basic Research Program | 202401AS070070 | Honglei Zhang |
| Yunnan Basic Research Program | 202301AY070001-106 | Chao Liu |
| Yunnan Young and Middle-aged Academic and Technical Leaders Reserve Talents Project | 202005AC160048 | Honglei Zhang |
| Yunnan Provincial Department of Education Science Research Fund Project | 2024J0246 | Shuting Wang |
| "Famous Doctor" Special Project of Ten Thousand People Plan of Yunnan Province | CZ0096 | Gaofeng Li |
| "Famous Doctor" Special Project of Ten Thousand People Plan of Yunnan Province | YNWR-MY-2020-095 | Zhiyong Deng |
| Medical Leading Talents Training Program of Yunnan Provincal Health Commission | L-2019028 | Gaofeng Li |

The funders had no role in study design, data collection and interpretation, or the decision to submit the work for publication.

## Author contributions

Honglei Zhang, Conceptualization, Data curation, Supervision, Investigation, Writing – original draft, Project administration, Writing – review and editing; Chao Liu, Resources; Shuting Wang, Minjun Zhou, Investigation; Qing Wang, Huawei Jiang, Data curation; Xu Feng, Formal analysis; Li Xiao, Resources, Data curation, Investigation; Chao Luo, Data curation, Formal analysis, Funding acquisition; Lu Zhang, Resources, Formal analysis, Funding acquisition; Fei Hou, Xiaosan Su, Gaofeng Li, Resources, Funding acquisition, Investigation; Zhiyong Deng, Resources, Data curation, Formal analysis; Heng Li, Formal analysis, Investigation; Yong Zhang, Data curation, Funding acquisition, Investigation

## Author ORCIDs

Honglei Zhang http://orcid.org/0000-0002-5522-8120
Yong Zhang https://orcid.org/0000-0003-0513-8941
Xiaosan Su http://orcid.org/0000-0002-6163-2429

## Ethics

The study protocol was reviewed and approved by the ethical committees of Yunnan Cancer Hospital & The Third Affiliated Hospital of Kunming Medical University (KYCS2022067) and conformed to the ethical standards for medical research involving human subjects, as laid out in the 1964 Declaration of Helsinki and its later amendments.

Reviewer #1 (Public review): https://doi.org/10.7554/eLife.95453.3.sa1
Reviewer #2 (Public review): https://doi.org/10.7554/eLife.95453.3.sa2
Author response https://doi.org/10.7554/eLife.95453.3.sa3

---

# Additional files

## Supplementary files

• Supplementary file 1. Clinical information for TSLC, TNLC, CNLC, and XWLC cohorts. (**a**) Clinical data for TSLC and TNLC cohorts. (**b**) Clinical data for CNLC cohort. (**c**) Clinical data for XWLC

cohort.

• Supplementary file 2. Somatic mutation profile in XWLC. (**a**) Somatic mutation profile in XWLC. (**b**) Mutation frequency of cancer driver genes XWLC cohort.

• Supplementary file 3. Copy number variation results in XWLC.

• Supplementary file 4. Normalized mRNA expression (CPM values derived from EdgeR) in XWLC.

• Supplementary file 5. Proteomic expression results normalized by column (patient) median.

• Supplementary file 6. Quantitative phosphoproteomic data at phosphosite level normalized by column (patient) median.

• Supplementary file 7. Estimation of Immune cell infiltration with ssGSEA method.

• Supplementary file 8. Radiomics features in XWLC cohort.

• Supplementary file 9. Onco_PPIs identified in four cohorts. (a) onco_PPIs in XWLC cohort; (b) onco_PPIs in CNLC cohort; (c) onco_PPIs in TNLCF cohort; (d) onco_PPIs in TSLC cohort.

• MDAR checklist

### Data availability

Raw sequencing data have been deposited in the Genome Sequence Archive for Human (GSA-Human, https://ngdc.cncb.ac.cn/gsa-human/) under accession codes HRA000124, HRA001481 and HRA001482. Proteomics and phosphoproteomics data have been deposited in the Open Archive for Miscellaneous Data (OMIX, https://ngdc.cncb.ac.cn/omix/) under accession codes OMIX001292. The raw lung CT images used in this paper are available from OMIX under accession codes OMIX002491.

The following datasets were generated:

| Author(s) | Year | Dataset title | Dataset URL | Database and Identifier |
|---|---|---|---|---|
| Zhang H, Liu C, Wang S | 2021 | XWFA | https://ngdc.cncb.ac.cn/gsa-human/browse/HRA000124 | NGDC GSA for Human, HRA000124 |
| Zhang H, Liu C, Wang S | 2021 | XWFA_2nd_batch_WES | https://ngdc.cncb.ac.cn/gsa-human/browse/HRA001481 | NGDC GSA for Human, HRA001481 |
| Zhang H, Liu C, Wang S | 2021 | XWFA_2nd_batch_RNAseq | https://ngdc.cncb.ac.cn/gsa-human/browse/HRA001482 | NGDC GSA for Human, HRA001482 |
| Zhang H, Liu C, Zhang H | 2022 | LCCS proteomics | https://ngdc.cncb.ac.cn/omix/release/OMIX001292 | NGDC OMIX, OMIX001292 |
| Zhang H, Liu C, Wang S | 2022 | lung cancer imaging | https://ngdc.cncb.ac.cn/omix/release/OMIX002491 | NGDC OMIX, OMIX002491 |

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
