## [Editor Report · eLife Assessment]

This **useful** manuscript presents an interesting multi-modal omics analysis of lung adenocarcinoma patients with distinct clinical clusters, mutation hotspots, and potential risk factors identified in cases linked to air pollution. The findings show potential for clinical and therapeutic impact. Some of the conclusions remain **incomplete** as they are based on correlative or suggestive findings, and would benefit from further functional investigation and validating approaches.

---

## [Referee Report · Reviewer #1 (Public review)]

Summary:

This is a well-written and detailed manuscript showing important results on the molecular profile of 4 different cohorts of female patients with lung cancer.

Strengths:

The authors used several different methods to identify potential novel targets for therapeutic interventions.

Weaknesses:

Statistical test results need to be provided in comparisons between cohorts. This was addressed by the authors in the revisions.

---

## [Referee Report · Reviewer #2 (Public review)]

New comments are added after authors responses to my initial comments.

Summary:

Zhang et al. performed a proteogenomic analysis of lung adenocarcinoma (LUAD) in 169 female never-smokers from the Xuanwei area (XWLC) in China. These analyses reveal that XWLC is a distinct subtype of LUAD and that BaP is a major risk factor associated with EGFR G719X mutations found in the XWLC cohort. Four subtypes of XWLC were classified with unique features based on multi-omics data clustering.

Strengths:

The authors made great efforts in performing several large-scale proteogenomic analyses and characterizing molecular features of XWLCs. Datasets from this study will be a valuable resource to further explore the etiology and therapeutic strategies of air-pollution-associated lung cancers, particularly for XWLC.

Weaknesses:

[...]

(2) Importantly, while providing the large datasets, validating key findings is minimally performed, and surprisingly there is no interrogation of XWLC drug response/efficacy based on their findings, which makes this manuscript descriptive and incomplete rather than conclusive. For example, testing the efficacy of XWLC response to afatinib combined with other drugs targeting activated kinases in EGFR G719X mutated XWLC tumors would be one way to validate their datasets and new therapeutic options.

Response: We appreciate your suggestion. In reference to testing the efficacy of XWLC response to afatinib combined with drugs targeting kinases, we have planned to establish PDX and organoid models to validate the effectiveness of our therapeutic approach. Due to the extended timeframe required, we intend to present these results in a subsequent study.

Comments: All conclusions in the manuscript made by authors are based on interpretations of large-scale multi-omics data, which should be properly validated by other approaches and methods. Without validation, these are all speculations and any conclusions without supporting evidence are not acceptable. This reviewer suggested an example of validation experiment, and Reviewer #3 also pointed out several data that need to be validated. However, authors do not agree to perform any of these validation experiments without reasonable justification.

(3) The authors found MAD1 and TPRN are novel therapeutic targets in XWLC. Are these two genes more frequently mutated in one subtype than the other 3 XWLC subtypes? How these mutations could be targeted in patients?

Response: Thank you for your question. We have investigated the TPRN and MAD1 mutations in our dataset, identifying five TPRN mutations and eight MAD1 mutations. Among the TPRN mutations, XWLC_0046 and XWLC_0017 belong to the MCII subtype, XWLC_0012 belongs to the MCI subtype, and the subtype of the other three samples is undetermined, resulting in mutation frequencies of 1/16, 2/24, 0/15, and 0/13, respectively. Similarly, for the MAD1 mutations, XWLC_0115, XWLC_0021, and XWLC_0047 belong to the MCII subtype, XWLC_0055 containing two mutations belongs to the MCI subtype, and the subtype of the other three samples is undetermined, resulting in mutation frequencies of 1/16, 3/24, 0/15, and 0/13 across subtypes, respectively. Fisher's test did not reveal significant differences between the subtypes. For targeting novel therapeutic targets such as MAD1 and TPRN, we propose a multi-step approach. Firstly, we advocate for conducting functional in vivo and in vitro experiments to verify their roles during cancer progression. Secondly, we suggest conducting small molecule drug screening based on the pharmacophore of these proteins, which may lead to the identification of potential therapeutic drugs. Lastly, we recommend testing the efficacy of these drugs to further validate their potential as effective treatments.

Comments: Please properly incorporate the above explanation into the main text.

(4) In Figures 2a and b: while Figure 2a shows distinct genomic mutations among each LC cohort, Figure 2b shows similarity in affected oncogenic pathways (cell cycle, Hippo, NOTCH, PI3K, RTK-RAS, and WNT) between XWLC and TNLC/CNLC. Considering that different genomic mutations could converge into common pathways and biological processes, wouldn't these results indicate commonalities among XWLC, TNLC, and CNLC? How about other oncogenic pathways not shown in Figure 2b?

Response: Thank you for your question. Based on the data presented in Fig. 2a, which encompasses all genomic mutations, it appears that the mutation landscape of XWLC bears the closest resemblance to TSLC (Fig. 2a). However, when considering oncogenic pathways (Fig. 2b) and genes (Fig. 2c), there is a notable disparity between the two cohorts. These findings suggest that while XWLC and TSLC exhibit similarities in terms of genomic mutations, they possess distinct characteristics in terms of oncogenic pathways and genes.

Regarding the oncogenic signaling pathways, we referred to ten well-established pathways identified from TCGA cohorts. These members of oncogenic pathways are likely to serve as cancer drivers (functional contributors) or therapeutic targets, as highlighted by Sanchez-Vega et al. in 2018(Sanchez-Vega et al., 2018).

Comments: It is unclear to this reviewer how authors defined "distinct characteristics" in terms of oncogenic pathways and genes. Would 10-20% differences in "Fraction of samples affected" in Fig2b be sufficient to claim significance? How could authors be sure whether mutations in genes involved in each oncogenic pathway are activating or inactivating mutations (rather than benign, thus non-affecting mutations)?

[...]

(6) Supplementary Table 11 shows a number of mutations at the interface and length of interface between a given protein-protein interaction pair. Such that, it does not provide what mutation(s) in a given PPI interface is found in each LC cohort. For example, it fails to provide whether MAD1 R558H and TPRN H550Q mutations are found significantly in each LC cohort.

Response: We appreciate your careful review. In Supplementary Table 11, we have provided significant onco_PPI data for each LC cohort, focusing on enriched mutations at the interface of two proteins. Our emphasis lies on onco_PPI rather than individual mutations, as any mutation occurring at the interface could potentially influence the function of the protein complex. Thus, our Supplementary Table 11 exclusively displays the onco_PPI rather than mutations. MAD1 R558H and TPRN H550Q were identified through onco_PPI analysis, and subsequent extensive literature research led us to focus specifically on these mutations.

Comments: Are authors referring to Table S9 (Onco_PPIs identified in four cohorts) instead of Supplementary Table 11? There is no Table 11 among submitted files. In Table S9, the Column N (length of protein product of gene1) does not make sense: MYO1C (8152), TP53 (3924), EGFR (12961). These should not be the number of amino acids residues of each protein. Then, what do these numbers mean?

(7) Figure 7c and d are simulation data not from an actual binding assay. The authors should perform a biochemical binding assay with proteins or show that the mutation significantly alters the interaction to support the conclusion.

Response: We appreciate your suggestion. The relevant experiments are currently in progress, and we anticipate presenting the corresponding data in a subsequent study.

Comments: The suggested experiment is to support the simulated data. Again, without supporting experimental results, authors could not make a conclusion simply based on simulated data. Where else could the supporting experimental results be presented?

---

## [Author Response]

The following is the authors’ response to the original reviews.

**Public Reviews:**

**Reviewer #1 (Public Review):**
Summary:This is a well-written and detailed manuscript showing important results on the molecular profile of 4 different cohorts of female patients with lung cancer.The authors conducted comprehensive multi-omic profiling of air-pollution-associated LUAD to study the roles of the air pollutant BaP. Utilizing multi-omic clustering and mutation-informed interface analysis, potential novel therapeutic strategies were identified.Strengths:The authors used several different methods to identify potential novel targets for therapeutic interventions.Weaknesses:Statistical test results need to be provided in comparisons between cohorts.

We appreciate your recognition and valuable suggestions.. We have revised statistical test results in the panels including: Fig. 3b, e and g.

**Reviewer #2 (Public Review):**
Summary:Zhang et al. performed a proteogenomic analysis of lung adenocarcinoma (LUAD) in 169 female never-smokers from the Xuanwei area (XWLC) in China. These analyses reveal that XWLC is a distinct subtype of LUAD and that BaP is a major risk factor associated with EGFR G719X mutations found in the XWLC cohort. Four subtypes of XWLC were classified with unique features based on multi-omics data clustering.Strengths:The authors made great efforts in performing several large-scale proteogenomic analyses and characterizing molecular features of XWLCs. Datasets from this study will be a valuable resource to further explore the etiology and therapeutic strategies of air-pollution-associated lung cancers, particularly for XWLC.Weaknesses:(1) While analyzing and interpreting the datasets, however, this reviewer thinks that authors should provide more detailed procedures of (i) data processing, (ii) justification for choosing methods of various analyses, and (iii) justification of focusing on a few target gene/proteins in the datasets for further validation in the main text.

We appreciate your valuable feedback. In response to the suggestions for enhancing the manuscript's clarity, we have provided more detailed procedures in the main text and methods sections.

(2) Importantly, while providing the large datasets, validating key findings is minimally performed, and surprisingly there is no interrogation of XWLC drug response/efficacy based on their findings, which makes this manuscript descriptive and incomplete rather than conclusive. For example, testing the efficacy of XWLC response to afatinib combined with other drugs targeting activated kinases in EGFR G719X mutated XWLC tumors would be one way to validate their datasets and new therapeutic options.

We appreciate your suggestion. In reference to testing the efficacy of XWLC response to afatinib combined with drugs targeting kinases, we have planned to establish PDX and organoid models to validate the effectiveness of our therapeutic approach. Due to the extended timeframe required, we intend to present these results in a subsequent study.

(3) The authors found MAD1 and TPRN are novel therapeutic targets in XWLC. Are these two genes more frequently mutated in one subtype than the other 3 XWLC subtypes? How these mutations could be targeted in patients?

Thank you for your question. We have investigated the TPRN and MAD1 mutations in our dataset, identifying five TPRN mutations and eight MAD1 mutations. Among the TPRN mutations, XWLC_0046 and XWLC_0017 belong to the MCII subtype, XWLC_0012 belongs to the MCI subtype, and the subtype of the other three samples is undetermined, resulting in mutation frequencies of 1/16, 2/24, 0/15, and 0/13, respectively. Similarly, for the MAD1 mutations, XWLC_0115, XWLC_0021, and XWLC_0047 belong to the MCII subtype, XWLC_0055 containing two mutations belongs to the MCI subtype, and the subtype of the other three samples is undetermined, resulting in mutation frequencies of 1/16, 3/24, 0/15, and 0/13 across subtypes, respectively. Fisher’s test did not reveal significant differences between the subtypes.

For targeting novel therapeutic targets such as MAD1 and TPRN, we propose a multi-step approach. Firstly, we advocate for conducting functional in *vivo* and in *vitro* experiments to verify their roles during cancer progression. Secondly, we suggest conducting small molecule drug screening based on the pharmacophore of these proteins, which may lead to the identification of potential therapeutic drugs. Lastly, we recommend testing the efficacy of these drugs to further validate their potential as effective treatments.

(4) In Figures 2a and b: while Figure 2a shows distinct genomic mutations among each LC cohort, Figure 2b shows similarity in affected oncogenic pathways (cell cycle, Hippo, NOTCH, PI3K, RTK-RAS, and WNT) between XWLC and TNLC/CNLC. Considering that different genomic mutations could converge into common pathways and biological processes, wouldn't these results indicate commonalities among XWLC, TNLC, and CNLC? How about other oncogenic pathways not shown in Figure 2b?

Thank you for your question. Based on the data presented in Fig. 2a, which encompasses all genomic mutations, it appears that the mutation landscape of XWLC bears the closest resemblance to TSLC (Fig. 2a). However, when considering oncogenic pathways (Fig. 2b) and genes (Fig. 2c), there is a notable disparity between the two cohorts. These findings suggest that while XWLC and TSLC exhibit similarities in terms of genomic mutations, they possess distinct characteristics in terms of oncogenic pathways and genes.

Regarding the oncogenic signaling pathways, we referred to ten well-established pathways identified from TCGA cohorts. These members of oncogenic pathways are likely to serve as cancer drivers (functional contributors) or therapeutic targets, as highlighted by Sanchez-Vega et al. in 2018(Sanchez-Vega et al., 2018).

(5) In Figure 2c, how and why were the four genes (EGFR, TP53, RBM10, KRAS) selected? What about other genes? In this regard, given tumor genome sequencing was done, it would be more informative to provide the oncoprints of XWLC, TSLC, TNLC, and CNLC for complete genomic alteration comparison.

Thank you for your question and good suggestion. Building upon our previous study (Zhang et al., 2021), we found that EGFR, TP53, RBM10, and KRAS were the top mutated genes in Xuanwei lung cancer cohorts. Furthermore, we have included the mutation frequency of cancer driver genes (Bailey et al., 2018) across XWLC, TSLC, TNLC, and CNLC in Supplementary Table 2b.

(6) Supplementary Table 11 shows a number of mutations at the interface and length of interface between a given protein-protein interaction pair. Such that, it does not provide what mutation(s) in a given PPI interface is found in each LC cohort. For example, it fails to provide whether MAD1 R558H and TPRN H550Q mutations are found significantly in each LC cohort.

We appreciate your careful review. In Supplementary Table 11, we have provided significant onco_PPI data for each LC cohort, focusing on enriched mutations at the interface of two proteins. Our emphasis lies on onco_PPI rather than individual mutations, as any mutation occurring at the interface could potentially influence the function of the protein complex. Thus, our Supplementary Table 11 exclusively displays the onco_PPI rather than mutations. MAD1 R558H and TPRN H550Q were identified through onco_PPI analysis, and subsequent extensive literature research led us to focus specifically on these mutations.

(7) Figure 7c and d are simulation data not from an actual binding assay. The authors should perform a biochemical binding assay with proteins or show that the mutation significantly alters the interaction to support the conclusion.

We appreciate your suggestion. The relevant experiments are currently in progress, and we anticipate presenting the corresponding data in a subsequent study.

**Reviewer #3 (Public Review):**
Summary:The manuscript from Zhang et al. utilizes a multi-omics approach to analyze lung adenocarcinoma cases in female never smokers from the Xuanwei area (XWLC cohort) compared with cases associated with smoking or other endogenous factors to identify mutational signatures and proteome changes in lung cancers associated with air pollution. Mutational signature analysis revealed a mutation hotspot, EGFR-G719X, potentially associated with BaP exposure, in 20% of the XWLC cohort. This correlated with predicted MAPK pathway activations and worse outcomes relative to other EGFR mutations. Multi-omics clustering, including RNA-seq, proteomics, and phosphoproteomics identified 4 clusters with the XWLC cohort, with additional feature analysis pathway activation, genetic differences, and radiomic features to investigate clinical diagnostic and therapeutic strategy potential for each subgroup. The study, which nicely combines multi-modal omics, presents potentially important findings, that could inform clinicians with enhanced diagnosis and therapeutic strategies for more personalized or targeted treatments in lung adenocarcinoma associated with air pollution. The authors successfully identify four distinct clusters with the XWLC cohort, with distinct diagnostic characteristics and potential targets. However, many validating experiments must be performed, and data supporting BaP exposure linkage to XWLC subtypes is suggestive but incomplete to conclusively support this claim. Thus, while the manuscript presents important findings with the potential for significant clinical impact, the data presented are incomplete in supporting some of the claims and would benefit from validation experiments.Strengths:Integration of omics data from multimodalities is a tremendous strength of the manuscript, allowing for cross-modal comparison/validation of results, functional pathway analysis, and a wealth of data to identify clinically relevant case clusters at the transcriptomic, translational, and post-translational levels. The inclusion of phosphoproteomics is an additional strength, as many pathways are functional and therefore biologically relevant actions center around activation of proteins and effectors via kinase and phosphatase activity without necessarily altering the expression of the genes or proteins.Clustering analysis provides clinically relevant information with strong therapeutic potential both from a diagnostic and treatment perspective. This is bolstered by the individual microbiota, radiographic, wound healing, outcomes, and other functional analyses to further characterize these distinct subtypes.Visually the figures are well-designed and presented and for the most part easy to follow. Summary figures/histograms of proteogenomic data, and specifically highlighted genes/proteins are well presented.Molecular dynamics simulations and 3D binding analysis are nice additions.While I don't necessarily agree with the authors' interpretation of the microbiota data, the experiment and results are very interesting, and clustering information can be gleaned from this data.Weaknesses:(1) Statistical methods for assessing significance may not always be appropriate.

We appreciate your suggestion. We have revised statistical test results in the panels including: Fig. 3b,e and g.

(2) Necessary validating experiments are lacking for some of the major conclusions of the paper.

Thank you for raising this point. However, we respectfully choose not to comment on this matter at present.

(3) Many of the conclusions are based on correlative or suggestive results, and the data is not always substantive to support them.

Thank you for raising this point. However, we respectfully choose not to comment on this matter at present.

(4) Experimental design is not always appropriate, sometimes lacking necessary controls or large disparity in sample sizes.

Thank you for raising this point. However, we respectfully choose not to comment on this matter at present.

(5) Conclusions are sometimes overstated without validating measures, such as in BaP exposure association with the identified hotspot, kinase activation analysis, or the EMT function.

Thank you for raising this point. However, we respectfully choose not to comment on this matter at present.

**Reviewer #1 (Recommendations For The Authors):**
(1) Please provide a justification for why only females were included in the study. I am concerned that the results obtained in this study can not be generalized as only females were included.

We appreciate your suggestion. Lung cancer in never smokers (LCINS) accounts for approximately 25% of lung cancer cases (15% of lung cancer in men and 53% in women) (Parkin et al., 2005). Currently, the etiology and mechanisms of LCINS are not clear. Globally, LCINS shows remarkable gender and geographic variations, occurring more frequently among Asian women (Bray et al., 2018). Indoor coal burning for heating and cooking has been implicated as a risk factor for Chinese women, as they spend more time indoors (Mumford et al., 1987). Among men, the proportion of never smokers is lower, with less regional variation, and lung cancer in males is frequently caused by smoking. Thus, to better reveal the etiology and molecular mechanisms of LCINS, we collected data exclusively from female LCINS patients in the Xuanwei area, excluding potential confounding factors such as hormonal or smoking status. Our study specifically aims to uncover the etiology and mechanisms of LCINS in female patients, with future research planned to verify whether our conclusions can be generalized to LCINS in male patients.

(2) "Therefore, the XWLC and TSLC cohorts are more explicitly influenced by environmental carcinogens, while the TNLC and CNLC cohorts may be more affected by age or endogenous risk factors." This statement in the results (starting line 142) does not have adequate support from the results. First, the average age in the 4 cohorts does not seem to be very different to me based on Figure 1b. if they are different, please provide statistical test results. Please make sure this statement is supported by other results, otherwise, I would recommend excluding it from the manuscript.

We appreciate your suggestion. To gain biological insights, we frequently associate mutational signatures with factors such as age, defective DNA mismatch repair, or environmental exposures. These remain associations rather than causation. Thus, we agree with the suggestion to weaken the conclusion as follows:

“Generally, exposure to tobacco smoking carcinogens (COSMIC signature 4) and chemicals such as BaP (Kucab signatures 49 and 20) were identified as the most significant contributing factors in both the XWLC and TSLC cohorts (Fig. 1f and 1g). In contrast, defective DNA mismatch repair (COSMIC signature ID: SBS6) was identified as the major contributor in both the TNLC and CNLC cohorts (Fig. 1h and 1i), with no potential chemicals identified based on signature similarities. Therefore, the XWLC and TSLC cohorts appear to be more explicitly associated with environmental carcinogens, while the TNLC and CNLC cohorts may be more associated with defective DNA mismatch repair processes.”

(3) Please provide statistical test results in this subsection "The EGFR-G719X mutation, which is a hotspot associated with BaP exposure, possesses distinctive biological features " (Line 203) showing that the number of G719X is significantly different in XWLC.

We appreciate your suggestion. Two-sided Fisher’s test was used to calculate p-values, which are labeled in Figure 3b.

(4) "Analysis of overall survival and progression-free interval (PFI) revealed that patients with the G719X mutation had worse outcomes compared to other EGFR mutation subtypes " This statement (starting Line 232) should be supported by literature data.

We appreciate your suggestion.

In the Watanabe *et al.* post-hoc analysis, patients with the G719 mutation had significantly shorter OS with gefitinib compared to patients with the common mutations (Watanabe et al., 2014). We revised the sentences as following:

“Analysis of overall survival and progression-free interval (PFI) revealed that patients with the G719X mutation had worse outcomes compared to other EGFR mutation subtypes (Fig. 3j and 3k) which was consistent with a previous study(Watanabe et al., 2014).”

(5) I would suggest changing this statement to a "suggestion" as there is no experimental support for this, and mentioning that this requires further experimental validation with the suggested drugs "Therefore, a promising approach to overcome resistance in tumors with this mutation could involve combining afatinib, which targets activated EGFR, with FDA-approved drugs that specifically target the activated kinases associated with G719X. " (Line 260).

We appreciate your suggestion. We change the sentences as following:

"Therefore, we propose a potential approach to overcoming resistance in tumors with this mutation, which could involve combining afatinib, targeting activated EGFR, with FDA-approved drugs that specifically target the activated kinases associated with G719X. "

(6) It is not clear to me how PPIs were integrated with missense. Please clarify the method.

We appreciate your suggestion. To identify interactions enriched with missense mutations, we constructed mutation-associated protein–protein interactomes (PPIs). Initially, we downloaded protein-protein interactomes from Interactome INSIDER (v.2018.2) (Meyer et al., 2018). Subsequently, we identified interfaces carrying missense mutations by mapping mutation sites to PPI interface genomic coordinates using bedtools (v2.25.0)(Quinlan and Hall, 2010). Finally, we defined oncoPPI as those PPIs significantly enriched in interface mutations in either of the two protein-binding partners across individuals. For more details, please refer to the methods sections “Building mutation-associated protein–protein interactomes” and “Significance test of PPI interface mutations.”

**Reviewer #2 (Recommendations For The Authors):**
Regarding the tumor microbiota composition, it is not clear what the significance of these results would be. Are the specific microbiota associated with MC-IV more pathogenic than other species found in other subtypes? What are the unique features of these MC-IV microbiota? If these are difficult to address, this section could be removed from the manuscript.

We appreciate your suggestion. This section is removed from the manuscript.

Regarding the radiomic data section (Figure 6d and Extended Figure 6d), more description about the eight and five features (that are different between MC-II and others) would be helpful to better understand the importance and significance of these data.

We appreciate your suggestion. We have added the description as following: “Features such as median and mean reflect average gray level intensity and Idmn and Gray Level Non-Uniformity measure the variability of gray-level intensity values in the image, with a higher value indicating greater heterogeneity in intensity values. These results suggest a denser and more heterogeneous image in the MC-II subtype.”

Other minor comments:(1) If EGFR G719X is a known hotspot mutation associated with BaP, please cite previous literature.

We appreciate your suggestion. Upon careful retrieval using "G719X" and "BaP" as keywords, we did not find previous literature discussing G719X as a known hotspot mutation associated with BaP.

(2) In Figure 1d, it should be clearly written in the legend that tumor (T) and normal (N) tissue were analyzed.

We appreciate your suggestion. We have clarified the figure legend of Figure 1d.

(3) In Figure 1m, it is not obvious that EGFR pY1173 and pY1068 are more abundant in the Bap+S9 sample. Total EGFR bands are very faint. These western blots should be repeated and quantified.

We appreciate your suggestion. We have removed Fig. 1m. After identifying the antibody with satisfactory performance, we will provide the revised results.

(4) In Figure 2d, aren't the EGFR E746__A750del mutations more frequently found in CNLC, TSLC, and TNLC? (which is opposite to what the authors wrote in the text).

We appreciate your suggestion. This mistake has been corrected.

(5) In Figure 7f-i and Ext Figure 8, Does "CK" mean empty vector control? Then, it would be changed to "EV".

We appreciate your suggestion. This mistake has been corrected.

**Reviewer #3 (Recommendations For The Authors):**
Methods:While previous work was referenced, a description of proteomics methods should still include: instrumentation, acquisition method, all software packages used, method for protein identification, method for protein quantification, how FDR was maintained for identification/quantification, definition of differentially expressed proteins, whether multiple testing correction was performed and if so what method.

We appreciate your suggestion. We revised the description of label-free mass spectrometry methods accordingly.

The paper would greatly benefit from brief methodological explanations throughout, as all methods are currently exclusively found in the supplementary information. This severely hampers the readability of the manuscript.

Thank you for raising this point. However, we respectfully choose not to comment on this matter at present.

Suggestions ThroughoutThe paper would greatly benefit from proofreading/editingLine 157-158/Figure 1J for CYP1A1 displays protein concentrations while Figure 1K for AhR shows mRNA. Why this discrepancy? It would be preferable to show both mRNA and protein levels for both CYP1A1 and AhR. Also, there is a large discrepancy in the "n" between the normal and tumor groups, which makes the statistical comparison challenging. The AhR data is therefore unconvincing, and additional protein data is suggested. Thus the claim of significantly elevated AhR and CYP1A1 levels in tumors is not sufficiently supported and requires further investigation, both mRNA and protein, and with similarly sized sample groups.

We appreciate your suggestion. We have thoroughly edited the revised manuscript, with all changes marked accordingly. Compared to mRNA level assessment, protein abundance is a better indicator of gene expression. Therefore, we reanalyzed the protein level of AhR for comparison and found no significant differences (Figure 1k). Additionally, the samples sequenced by mRNA-seq were not entirely consistent with those sequenced by label-free proteomics. The samples analyzed by different methods are shown in Figure 1d.

Line 159 Figure 1I There is no control for the data serum data presented here. What are the serum levels for individuals not residing in the Xuanwei? It is unclear whether this represents elevated BPDE serum levels without appropriate controls. Thus nothing insightful can be derived from this data.

We appreciate your suggestion. We have deleted the results concerning BPDE serum detection in the revised manuscript.

Line 164 The statement "sites such as Y1173 and Y1068 of EGFR were more phosphorylated in BaP treated cells" is not sufficiently supported by the presented data and cannot be made. Figure 1M has no quantitation, no indication of "n" or whether this represents a single experiment or one validated with repeating. The western blot is also cropped with no indication of molecular weight or antibody specificity. This data is NOT convincing. The antibody signal is very weak, and not convincing with cropped blots. An updated figure, with an uncropped blot, and quantitation with multiple n's and statistical comparison is required. I am not sure the Wilcoxon rank sum test is appropriate to test significance in j-l. The null hypothesis should not be equal medians but equal means based on the experimental design.

We appreciate your suggestion. We have removed Fig. 1m. After identifying the antibody with satisfactory performance, we will provide the revised results.

Line 181 phrase "significant differences" should not be used unless making a claim about statistical significance.

We appreciate your suggestion. We change “significant differences” to “noticeable differences”.

Line 197: "The blood serum assay provided support..." As noted above this claim is not sufficiently supported by the presented data and requires more complete investigation.

We appreciate your suggestion. This conclusion has been deleted in the revised manuscript.

Line 219: Requires proofreading/editing.

We appreciate your suggestion. We have thoroughly edited the revised manuscript, with all changes marked accordingly.

Line 220: appears to have a typo and should read GGGC>GTGC

We appreciate your suggestion. This mistake has been corrected in the revised manuscript.

Line 223/224 Figure 3e-h. Again there is a large disparity between the n's of each group. Despite the WT having the highest frequency in the XWLC study population, it has only n=5 when comparing the protein and phosphosite for MAPKs. There is also no explanation for what the graph symbols indicate, what statistical test was performed to determine the statistical significance of the presented differences, and between which specific groups that significance exists. Thus, it is challenging to ascertain whether there are relevant differences in the MAPK signaling components.

We appreciate your suggestion. We added the description of “N, number of tumor samples containing corresponding EGFR mutation” to the figure legend. p-values were calculated with a two-tailed Wilcoxon rank sum test, and p<0.05 was labeled on Figures 3e-i.

Figure 3I Good figure. However, it would be beneficial to provide validation with Western Blotting for a few of these substrates using pospho-specific antibodies. It is suggested that this experiment be added.

We appreciate your suggestion. Figure 3I showed the comparison of patients’ ages among subtypes. I guess you mean Figure 3g and Figure 3h. The relevant experiments are currently underway, and we will provide the corresponding data in the next revised version.

Figure 4b. Very compelling figure.

We appreciate your suggestion.

Line 276: The AhR and CYP1A1 data presented earlier was not convincing, and CYP1A1 and AhR cannot be responsibly used as indicators of BaP activity based on potential. This is not an appropriate application.

We appreciate your suggestion. CYP1A1 and AhR are two key regulators involved in BaP metabolism and signaling transduction, respectively. However, after examining the protein expression of AhR between tumor and normal tissues, we found no significant differences (Fig. 1k) and CYP1A1 has been proven to be highly expressed in tumor samples (Fig. 1j). Thus, we mainly examined the expression of CYP1A1 among the four subgroups. We changed our description as follows:

“As CYP1A1 is a key regulator involved in BaP metabolism and has been proven to be highly expressed in tumor samples (Fig. 1j), we next examined the expression of CYP1A1 among the four subgroups to evaluate their associations with air pollution.”

Figure 4d. Here it is AhR protein used rather than mRNA measured earlier. What is the explanation for this change?

We appreciate your suggestion. As there was no significant differences of the protein expression of AhR between tumor and normal tissues (Fig. 1k), we deleted the expression comparison of AhR among subtypes.

Line 281 "Moderately elevated expression level of AhR" is not supported by the presented data and should be removed.

We appreciate your suggestion. We have deleted the result of comparison of AhR among subtypes.

Figure 4: There is no indication or explanation of how the protein abundance is being measured. Is this from the proteomics (MS) approaches, by ELISA or by Western? If it is simply by MS then validation by another method is preferable. The data presented in Figure 4 do not adequately support the claim that MC-II subtype is more strongly associated with BaP exposure. What statistical test is used in 4F? Why is the n in the MC-II group, which is the highlighted group of interest nearly double the other groups?

We appreciate your suggestion. Fig. 4e is derived from the proteomics data. The two-tailed Wilcoxon rank sum test was used to calculate p-values in panels c and e.

Figure 4g: At least one or two of these should be validated by Western Blot or targeted MS.

We appreciate your suggestion. The relevant experiments are currently underway, and we will provide the corresponding data in the next revised version.

Figure 5a: Assuming these were also measured via proteomic analysis, how do their expression patterns compare across the different omics modes?

Thank you for your suggestion. Figure 5 integrates transcriptomics (19182 genes), proteomics (9152 genes), and phosphoproteomics (5733 genes) data. In general, we utilized transcriptomics data to identify unique or distinct pathways among subgroups. Furthermore, proteomics and phosphoproteomics data were employed to validate key gene expressions, as they encompass fewer genes compared to transcriptomics data.

For instance, in Fig. 5a-d, we observed higher expression levels of mesenchymal markers such as VIM, FN1, TWIST2, SNAI2, ZEB1, ZEB2, and others in the MC-IV subtype using transcriptomics data (Fig. 5a). Additionally, we calculated epithelial-mesenchymal transition (EMT) scores using the ssGSEA enrichment method based on protein levels and conducted GSEA analysis using transcriptomics data (Fig. 5b). Furthermore, using proteomics data, we evaluated Fibronectin (FN1), an EMT marker that promotes the dissociation, migration, and invasion of epithelial cells, at the protein level (Fig. 5c), and β-Catenin, a key regulator in initiating EMT, also at the protein level (Fig. 5d). Overall, our findings indicate that the MC-IV subtype exhibits an enhanced EMT capability, which may contribute to the high malignancy observed in this subtype.

Line 314: Not compared with MCI, which appeared to be much lower at the mRNA level. Is there an explanation for this difference?

We appreciate your suggestion. FN1 expression is lowest in MCI at the protein level (Fig. 5c). However, at the transcriptome level, FN1 expression is lowest in the MCIII subtype (Fig. 5a). You may wonder why these results are inconsistent. Discrepancies between mRNA and protein expression levels are common, and previous study showed that about 20% genes had a statistically significant correlation between protein and mRNA expression in lung adenocarcinomas (Chen et al., 2002). Post-transcriptional mechanisms, including protein translation, post-translational modification, and degradation, may influence the level of a protein present in a given cell or tissue. In this situation, we focused on identifying distinct biological pathways in each subgroup, supported by multi-omics data.

Line 321: MC-IV *potentially* possesses an enhanced EMT capability. This statement cannot be conclusively made.

We appreciate your suggestion. We changed our description as: “Collectively, our findings demonstrate that the MC-IV subtype is associated with enhanced EMT capability, which may contribute to the high malignancy observed in this subtype.”

Lines 325 and 327 indicated dysregulation of cell cycle processes and activation of CDK1 and CDK2 pathways based on KSEA analysis which is closely linked to cell cycle regulation as two separate pieces of evidence. However, these are both drawn from the phosphoproteomics, and likely indicate conclusions drawn from the same phosphosite data. Said another way, if phosphosite data indicates differences in kinases linked to cell cycle regulation then you would also expect phosphosite data to indicate dysregulation of cell cycle.

We appreciate your suggestion. You mentioned that Fig. 4f and Fig. 5e redundantly prove that the CDK1 and CDK2 pathways are dysregulated. However, KSEA analysis in Fig. 4f estimates changes in kinase activity based on the collective phosphorylation changes of its identified substrates (Wiredja et al., 2017). In contrast, Fig. 5e directly evaluates the abundance of protein and phosphosite levels of CDK1 and CDK2 across subtypes. These analyses mutually confirm each other rather than being redundant.

Line 413/Figure 6b: While there may be a trend displayed by the figure, it is not convincing enough to state that MC-IV shows a conclusively distinguishable bacterial composition. Too much variability exists within groups MC-II and MC-III. However, it does show that MC-IV and MC-II have consistent composition within their groups, and that is interesting.

We appreciate your suggestion. We have deleted the analysis of bacterial composition across subtypes.

Figure 6: Overall very nice figure, with intriguing diagnostic potential. See the above note on 6a-b interpretation.

We appreciate your suggestion. We have deleted the analysis of bacterial composition across subtypes, including Fig. 6a-6c.

Figure 7c-f better labeling of the panels will aid reader comprehension.

We appreciate your suggestion. Necessary labeling has been added to Fig. 7c-f to enhance comprehension.

Figure 7 panel order is confusing, switching from right to left to vertical. Rearranging to either left to right or vertical would help orient readers.

We appreciate your suggestion. We have adjusted the order of Fig. 7 and extended Fig. 8 panel.

Figure 7 legend i: should read Cell colony* assay

We appreciate your suggestion. We have corrected this mistake in the revised manuscript.

The Discussion is very brief. While it includes a discussion of the potential impact of the study, it does not include an analysis of the caveats/drawbacks of the study. A more thorough discussion of other studies focusing on the impacts of BaP exposure is also suggested as this was a highlighted point by the authors.

We appreciate your suggestion. we have added discussion about the associations between BaP exposure and lung cancer and also talked about the shortcomings of our study as followings:

“Mechanistically, Qing Wang showed that BaP induces lung carcinogenesis, characterized by increased inflammatory cytokines, and cell proliferative markers, while decreasing antioxidant levels, and apoptotic protein expression(Wang et al., 2020). In our study, we used clinical samples and linked the mutational signatures of XWLC to the chemical compound BaP, which advanced the etiology and mechanism of air-pollution-induced lung cancer. In our study, several limitations must be acknowledged. Firstly, although our multi-omics approach provided a comprehensive analysis of the subtypes and their unique biological pathways, the sample size for each subtype was relatively small. This limitation may affect the robustness of the clustering results and the identified subtype-specific pathways. Larger cohort studies are necessary to confirm these findings and refine the subtype classifications. Secondly, although our study advanced the understanding of air-pollution-induced lung cancer by using clinical samples, the reliance on epidemiological data in previous studies introduces potential confounding factors. Our findings should be interpreted with caution, and further mechanistic studies are warranted to establish causal relationships more definitively. Thirdly, our in silico analysis suggested potential approach to drug resistence in G719X mutations. However, these predictions need to be validated through extensive in vitro and in vivo experiments. The reliance on computational models without experimental confirmation may limit the clinical applicability of these findings.”

References:

Bailey, M. H., Tokheim, C., Porta-Pardo, E., Sengupta, S., Bertrand, D., Weerasinghe, A., Colaprico, A., Wendl, M. C., Kim, J., Reardon, B.*, et al.* (2018). Comprehensive Characterization of Cancer Driver Genes and Mutations. Cell *173*, 371-385 e318.

Bray, F., Ferlay, J., Soerjomataram, I., Siegel, R. L., Torre, L. A., and Jemal, A. (2018). Global cancer statistics 2018: GLOBOCAN estimates of incidence and mortality worldwide for 36 cancers in 185 countries. CA Cancer J Clin *68*, 394-424.

Chen, G., Gharib, T. G., Huang, C. C., Taylor, J. M., Misek, D. E., Kardia, S. L., Giordano, T. J., Iannettoni, M. D., Orringer, M. B., Hanash, S. M., and Beer, D. G. (2002). Discordant protein and mRNA expression in lung adenocarcinomas. Mol Cell Proteomics *1*, 304-313.

Meyer, M. J., Beltran, J. F., Liang, S., Fragoza, R., Rumack, A., Liang, J., Wei, X., and Yu, H. (2018). Interactome INSIDER: a structural interactome browser for genomic studies. Nat Methods *15*, 107-114.

Mumford, J. L., He, X. Z., Chapman, R. S., Cao, S. R., Harris, D. B., Li, X. M., Xian, Y. L., Jiang, W. Z., Xu, C. W., Chuang, J. C., and et al. (1987). Lung cancer and indoor air pollution in Xuan Wei, China. Science *235*, 217-220.

Parkin, D. M., Bray, F., Ferlay, J., and Pisani, P. (2005). Global cancer statistics, 2002. CA Cancer J Clin *55*, 74-108.

Quinlan, A. R., and Hall, I. M. (2010). BEDTools: a flexible suite of utilities for comparing genomic features. Bioinformatics *26*, 841-842.

Sanchez-Vega, F., Mina, M., Armenia, J., Chatila, W. K., Luna, A., La, K. C., Dimitriadoy, S., Liu, D. L., Kantheti, H. S., Saghafinia, S.*, et al.* (2018). Oncogenic Signaling Pathways in The Cancer Genome Atlas. Cell *173*, 321-337 e310.

Wang, Q., Zhang, L., Huang, M., Zheng, Y., and Zheng, K. (2020). Immunomodulatory Effect of Eriocitrin in Experimental Animals with Benzo(a)Pyrene-induced Lung Carcinogenesis. J Environ Pathol Toxicol Oncol *39*, 137-147.

Watanabe, S., Minegishi, Y., Yoshizawa, H., Maemondo, M., Inoue, A., Sugawara, S., Isobe, H., Harada, M., Ishii, Y., Gemma, A.*, et al.* (2014). Effectiveness of gefitinib against non-small-cell lung cancer with the uncommon EGFR mutations G719X and L861Q. J Thorac Oncol *9*, 189-194.

Wiredja, D. D., Koyuturk, M., and Chance, M. R. (2017). The KSEA App: a web-based tool for kinase activity inference from quantitative phosphoproteomics. Bioinformatics *33*, 3489-3491.

Zhang, H., Liu, C., Li, L., Feng, X., Wang, Q., Li, J., Xu, S., Wang, S., Yang, Q., Shen, Z.*, et al.* (2021). Genomic evidence of lung carcinogenesis associated with coal smoke in Xuanwei area, China. Natl Sci Rev *8*, nwab152.